

# Performance evaluation of four cascade impactors for airborne UFP collection: Influence of particle type, concentration, mass and chemical nature

Elisabeth Eckenberger[1], Andreas Mittereder[2], Nadine Gawlitta[3+4], Jürgen Schnelle-Kreis[3+4], Martin Sklorz[4], Dieter Brüggemann[2], Ralf Zimmermann[3+4], Anke C. Nölscher[1]

[1] Bayreuth Center of Ecology and Environmental Research (BayCEER), University of Bayreuth, Germany
[2] Department of Engineering Thermodynamics and Transport Processes, University of Bayreuth, Germany
[3] Comprehensive Molecular Analytics (CMA), Helmholtz Munich, Germany
[4] Chair of Analytical Chemistry, Institute of Chemistry, University of Rostock, Germany

*Correspondence to*: Elisabeth Eckenberger (Elisabeth.eckenberger@uni-bayreuth.de), Anke C. Nölscher (Anke.noelscher@uni-bayreuth.de)

**Abstract.** Ultrafine particles (UFP) have aerodynamic diameters of 100 nm or less. As UFP potentially impact human and environmental health, their chemical composition is of interest. However, their small mass presents challenges to techniques originally developed for larger particles. Therefore, we conducted a comprehensive characterization and comparison of four cascade impactors suitable to separate and collect UFP, namely 120R MOUDI (Micro-Orifice Uniform Deposit Impactor), ultraMOUDI, ELPI (Electrical Low-Pressure Impactor), and PENS (Personal Nanoparticle Sampler), under controlled laboratory conditions and in a field application.

In the laboratory, we evaluated pressure drop, cut-off diameters, steepness of the cut-off curve, losses, particle bounce, and transmitted particle mass. We observed performance differences among the impactors due to design and test aerosol mixture variations, including salt particles (NaCl), simulated secondary organic aerosol (SimSOA), and soot (with cut-off diameters of 59-68 nm, 70-74 nm, and 102-116 nm, respectively, as determined by electromobility diameter). All impactors successfully separated UFP, with the best agreement in cut-off diameters for SimSOA, showing maximum deviations of about 4 nm. The cut-off curve was steeper for soot compared to SimSOA and NaCl. Pressure drops were measured at 260±1 hPa (PENS), 420±2 hPa (ultraMOUDI), 600±3 hPa (120R MOUDI), and 690±3 hPa (ELPI). Losses were assessed as maximum transmission in the ultrafine fraction at 30 nm, resulting in 83±8% for PENS, 77±8% for ultraMOUDI, 75±8% for 120R MOUDI, and 69±7% for ELPI. We identified two additional factors crucial for mass-based analysis of organic marker compounds: evaporation of semi-volatile compounds due to high pressure drop across the impactor and material addition from larger particles bouncing off upper stages. Bounce-off was influenced by particle number concentration in the sampled air and could be mitigated by applying a coating to the upper impaction plates.



In the field application, we deployed the four cascade impactors side-by-side under environmental conditions to sample urban
and semi-industrial air. We analyzed six markers representing typical UFP sources and various molecular properties using
HPLC-MS/FLD (high-performance liquid chromatography with mass spectrometry and fluorescence detection). The markers
included polycyclic aromatic hydrocarbons (PAHs) such as benzo[a]pyrene (BaP) and benzo[b]fluoranthene (BbF),
levoglucosan (Levo), pinic acid (PA), terpenylic acid (TA), and N-(1,3-dimethylbutyl)-N'-phenyl-p-phenylenediamine
(6PPD). The impactors showed the best agreement for the two PAHs. BaP had an average mass concentration of $175\pm25$ pg/m³
across all impactors and sampling days, but concentrations varied by about 29% higher or 30% lower when analyzed with the
PENS and the 120R MOUDI, respectively, indicating a maximum disagreement of nearly 60%. The PENS consistently
reported higher mass concentrations for all marker compounds compared to the other impactors. Potential reasons for this
include the effects of pressure drop on gas-particle partitioning of semi-volatile compounds and material addition from particle
bounce, despite the applied coating. Semi-volatile markers PA, TA, and Levo exhibited decreasing absolute deviations from
the average mass concentration with increasing pressure drop, suggesting comparably higher evaporation losses during
sampling with the ELPI and lower losses with the PENS. Marker mass concentrations increased with higher air concentrations,
correlating with increased absolute deviations, likely due to bounce-off adding mass from larger particles. This effect was
strongest for the PENS, followed by ultraMOUDI, ELPI, and 120R MOUDI.

Overall, this study demonstrates the impact of impactor design, operational conditions, and aerosol mixture on observed mass
concentrations of organic markers in airborne UFP. Our findings highlight the complexities of accurately separating, collecting,
and analyzing UFP mass. While all four impactors can sample UFP, they each have distinct strengths and limitations that must
be considered when comparing atmospheric UFP study results.






## 1 Introduction

The characterization of ultrafine particles (UFP) in the atmosphere can be a key for understanding the cause and impact of air pollution. UFP are defined as particulate matter with aerodynamic diameters of 100 nanometers and less. They have gained

considerable attention due to their potential impacts on human health and the environment (Kumar et al., 2014, 2021; Schwarz et al., 2023). Furthermore, airborne UFP are naturally linked to weather and climate, as they might alter the radiative budget of Earth directly, or indirectly when growing and activating to form cloud droplets (Junkermann et al., 2022). The size and the chemical composition of UFP, thus, are crucial properties for assessing their sources, atmospheric fate, and potential adverse effects.


Information about size and chemical composition of airborne particulate matter can be collected via impactors. Impactors allow separating particles in an air sample based on their aerodynamic diameter ($d_a$). The physical separation of particles within an impactor is based on the particle's inertia, which causes relatively smaller particles to continue moving when the airflow bends sharply, e.g. when forced through nozzles, orifices, or slits. For predicting particle impaction, the dimensionless Stokes

number (Stk) can be used which describes the ratio of particle response time to the transit time of the air flow (Baron and Willeke, 2011):

$$Stk = \frac{\rho_P C_c d_p^2 U}{9 \eta W} \tag{1}$$


Where $\rho_P$ is the particle density, $C_c$ represents the Cunningham slip correction factor, $d_p$ is the particle diameter, U is the average air velocity at the nozzle exit, $\eta$ is the viscosity of the air or gas, and W is the diameter of the round nozzles. Particles with a Stokes number significantly greater than 1 are likely to impact on the collection surfaces due to their inability to follow the rapidly changing direction of the air streams within the impactor. Conversely, particles with lower Stokes numbers remain

airborne, following the air streamlines through the impactor (Baron and Willeke, 2011).

Upon separation, the particles can be collected on the impaction plates or downstream of the impactor, when placing filter substrates, such as aluminium foils, PTFE, or quartz fibre filters. Particles caught in the substrate can be extracted and undergo chemical analysis to determine their composition (Canepari et al., 2010; Daher et al., 2011). To describe the separation and

collection efficiency, the dp50, a value at which particles will be collected with 50% efficiency is used. The dp50 is proportional to the square root of the Stokes number for a given impactor design and flow rate.



A stack of impactors, impaction and nozzle plate pairs, is a cascade impactor. Within the cascade impactor, the nozzles or slits become increasingly smaller, so that the air moves increasingly faster through these orifices. Large particles impact early, while smaller particles travel further through the impactor and are collected at later stages.. The journey of cascade impactors began in 1860 when researchers aimed to collect and study airborne particles for the first time in order to understand their relationship with diseases (Marple, 2004). Later, from 1920 to 1945, impactors were refined for studying industrial aerosols in mine environments, shifting the focus to the quantification of particle concentrations. In 1945, the May cascade impactor provided insights into particle size distribution and concentration (May, 1945), which was spurred by military studies on chemical dispersion. Post-1945, numerous impactor designs proliferated, with many being variations of the original May cascade impactor. Theoretical analyses and computational methods from 1970 onwards revolutionized the understanding of impactors' fluid and particle dynamics, enabling precise size classification. In the mid-20th century, further progress was made by pushing cut-off sizes to sub-µm ranges (Berner et al., 1979; Brink, 1958; Mitchell and Pilcher, 1959). One example is the Berner impactor, which allowed the collection of UFP achieving a cut size of 0.082 µm at the lowest stage through an ingenious pressure drop mechanism. This concept evolved with the electrical low-pressure impactor (ELPI), which harnessed low pressure to achieve even smaller particle cut sizes and introduced electrometer-based charged particle detection with a potential for real-time data acquisition (Keskinen et al., 1992; Marjamäki et al., 2000). Another approach for ultrafine cut-off sizes uses small nozzles, such as implemented for the micro-orifice uniform deposit impactor (MOUDI) which is capable of capturing particles as small as 0.056 µm (Marple et al., 1991). Finally, miniaturized impactors were developed to study personal exposure, such as the Personal Nanoparticle Sampler (PENS) (Tsai et al., 2012).

The challenge, particularly when it comes to the separation and collection of UFP from the atmosphere using cascade impactors, is unequivocally associated with the remarkably low mass of UFP. This low mass, as opposed to fine particulate matter ($PM_{2.5}$), presents a substantial hurdle in conducting gravimetric or chemical analysis. Nevertheless, it is an essential prerequisite for elucidating the origins of particulate matter and evaluating the potential health risks associated with its presence. Specifically, UFP collection with cascade impactors is sensitive to the following aspects:

(1) The **cut-off diameter** critically determines the threshold at which particles are classified as either fine or ultrafine. A systematic sampling of particles larger than dp50=100 nm, even if only slightly above the threshold, can disproportionately dominate the collected mass, leading to misleading results. Referring to the cut-off diameter as aerodynamic diameter is an abstract measure describing a sphere with 1 $g/cm^3$ density settling with the same velocity as the respective particle. Thus, varying both formfactor and mass of real atmosphere's irregular particles will vary their impaction behaviour and hence the cut-off diameter and its sharpness.

(2) The **loss** of UFP on walls or earlier stages could retain particulate material on surfaces other than the filter substrate. These losses include larger particles, which may settle due to sedimentation or be swept away by turbulences, and smaller particles, which can adhere to surfaces through diffusion.



(3) The **bounce-off** could impact UFP sampling in two ways: (1) Larger particles break into fragments and re-entrain falsely adding material to the filter substrate dedicated for UFP. (2) Target UFP bounce-off from the collection

substrate and are lost for analysis.

(4) Semi-volatile particle-bound components may **evaporate** from already collected particulate surfaces during the ongoing sampling due to a reduced pressure within the impactor and continuous ventilation. This can result in alterations of the chemical composition, unwanted mass loss or gain, and a hampered conclusion.

Therefore, in this study, we explored the cut-off diameter, potential losses, artefacts due to bounce-off and evaporation for four different cascade impactors, which are capable of separating and sampling UFP. We did this with the overall aim to identify and quantify organic marker compounds in the collected UFP. The four cascade impactors differed in design, flow rate, number of stages, among other factors. We expected, that these differences would significantly impact the abovementioned aspects and therefore lead to different results of analyzed mass concentrations of the selected organic marker compounds within the

collected UFP.

## 2. Methods

### 2.1 Impactors for UFP Sampling

For our comparison, we selected four cascade impactors for sampling atmospheric UFP. Some of these impactors required adjustments to make them suitable for achieving the final cut-off at 100 nm. For each impactor and for each measurement, the

aerosol flow was regulated between the impactor and its pump. The flow rate was determined with a Gilibrator 2 bubble flow meter and re-checked following each collection interval.

**Rotating 10-stage 120R MOUDI-II**: The original MOUDI was designed for industrial hygiene studies with sampling periods ranging from a few minutes to several hours (Marple et al. (1991)). It utilizes numerous micro-orifice nozzles, which serve to

reduce jet velocity, pressure drop, particle bounce, and re-entrainment, while enhancing collection efficiency and enabling a uniform deposition. The uniform-deposit prevents particle build-up and allows a greater collection mass without overloading. It is further supported by rotating the impaction plates relative to the nozzles while distributing the sampled particles over 47 mm diameter sampling substrates. Earlier models (MOUDI 110 and 115) achieve relative rotation between impaction and nozzle plates by rotating alternate impactor stages using external gears and hooks that mesh with an external drive shaft

powered by an electric gear motor. In contrast, the newest model (120R MOUDI-II, MSP Corp., Shoreview, MN, USA), has impaction plates that are directly mounted onto miniature stepper motors, which are housed within a chamber situated in the center of each stage. Therefore, the 120R MOUDI-II offers a clean flow path from the impaction plate to the next stage's nozzle plate. This design is expected to show relatively lower particle losses compared to previous models (Marple et al., 2014).



Here, we operated the 120R MOUDI-II at a flow rate of 30 L/min and without the tenth impaction nozzle and plate. It separated airborne particulate matter into nine fractions with nine stages having nominal cut diameters of 0.10, 0.18, 0.32, 0.56, 1.0, 1.8, 3.2, 5.6, and 10 µm. We equipped all upper stages with aluminium foils with a 47 mm diameter and an after-filter holder with a 47 mm quartz fiber filter (Whatman QM-H) for collection of UFP.

**Non-rotating 3-stage MOUDI (ultraMOUDI):** Due to the versatility and previous successful utilization of MOUDI impactors in various studies, a compact version of the impactor was developed for this study.  In contrast to the 10-stage 120R MOUDI-II, this reduced version has three non-rotating stages. The internal construction of these stages is similar to that of the models 110 and 115, while sharing the same inlet and outlet design. The three stages have cut-off sizes of 0.1, 1 and 2.5 µm. Here, we employed the ultraMOUDI (MSP Corp., Shoreview, MN, USA) with aluminium foils (47 mm diameter) on the

impaction plates. The after-filter holder was equipped with a 47 mm quartz fiber filter (Whatman QM-H). The flow rate through the impactor was 30 L/min. Due to its reduced size and weight, this impactor could be integrated in an automated sampler for independent long-term observations of UFP.

**ELPI:** The ELPI (Dekati Ltd. in Tampere, Finland), introduced by Keskinen et al. in 1992, addresses a significant limitation

in impactor measurements, which is the particularly long sampling times required for gravimetric analysis being typically 24 hr and longer (Keskinen et al., 1992). Instead of accumulating and weighing the filter substrates offline, particles are counted on the respective stages within the ELPI while sampling. The ELPI determines particle size distributions ranging from 30 nm to 10 µm. Its major components are a unipolar diode charger, a cascade impactor, and a multichannel electrometer. Sampled particles are first charged to a predetermined level. These charged particles are then introduced into the cascade impactor,

which classifies the aerosols based on their inertia and, consequently, their aerodynamic diameter. The multichannel electrometer simultaneously measures the charges carried by the collected particles to each stage, providing a measurement for the particle number concentration. The particle classification is achieved using a multi-jet impactor, where stage 1 has the smallest and stage 13 has the largest cut-off of the particle aerodynamic diameter (0.03, 0.06, 0.09, 0.17, 0.26, 0.40, 0.65, 1.0, 1.6, 2.5. 4.4, 6.8, and 10 µm). The jet orifices are symmetrically drilled in rings around the center of each stage. Stage 1 of the

impactor serves as a critical orifice, regulating the flow rate to 30 L/min and creating a low-pressure, which is thought to ensure the impaction of even the smallest particles (Marjamäki et al., 2000).

For this study, we extracted the cascade impactor component from the ELPI and considered it as a standalone impactor without the charger and electrometer. For the collection of UFP, we modified it further. The lowest stages along with their

corresponding collection plates, namely stages 1 and 2 with cut-off sizes 0.03 and 0.06 µm, respectively, were removed to achieve a final cut-off size of 0.09 µm at stage 3. Instead, placeholders were used to secure the impactor plates using the built-in tensioner, ensuring the appropriate spacing between the nozzle plates and collection plates. On upper stages, aluminium foil





filters (25mm, Dekati) were installed. For collection of UFP, a 37 mm quartz fiber filter (Whatman QM-H) was installed in the afterfilter holder.


**PENS**: The PENS (Haze Control System Inc., Taiwan) was recently designed by Tsai et al. (2012) as a three-part system for the collection of airborne particles. The first part is a respirable cyclone that separates particles larger than 4 µm in aerodynamic diameter. The second part is a micro-orifice impactor, where particles with sizes from 0.1 to 4 µm impact on an impaction plate. The final part consists of a filter holder containing a 37 mm quartz fiber filter, which collects UFP. Compared to the
previously described devices, the PENS is compact with dimensions of 107 mm in length and 44 mm in width, and a total weight of 240 g. This makes the PENS portable and useful for personal exposure studies (Young et al., 2013). The PENS Impactor used in this study was modified to separate the particles in the fractions 2.5 and 0.1 µm and operates at a sampling flow rate of 4 L/min. The impaction plate itself is not suitable to install a filter. For collecting UFP within the afterfilter, we utilized a 37 mm quartz fiber filter (Whatman QM-H).

**2.2 Overview of laboratory tests for evaluating the performance of the four impactors for UFP separation and collection**

For testing the four cascade impactors under controlled conditions, we conducted a series of laboratory tests. Our objective was to determine the pressure-drop, transmission, UFP cut-off, and potential artefacts of each impactor. Therefore, we generated three types of test aerosol and used reference instrumentation to quantify UFP. This way we detected the particle number size distribution before and after passing the test aerosol through the impactors.

**2.2.1 Detection of particle number size distributions**

We monitored the particle number size distribution for the four cascade impactors with two complementing instruments. For accurate and sensitive observations, a mobility particle size spectrometer (MPSS) was deployed. However, the analysis of cut-off performance for impactors with a pressure drop of several hundred hectopascal (hPa) posed a challenge to the MPSS flow regulation. Thus, for fast and repetitive measurements in low pressure, a real-time differential mobility particle spectrometer
(DMS) was used. Yet, this instrument has comparatively low sensitivity.

**MPSS**

The MPSS (TROPOS) classifies and quantifies airborne particles based on the principle of the mobility of charged particles in an electric field (Wiedensohler et al., 2012a, 2018). Particles contained in sampled air are initially neutralized using a
radioactive source to achieve a Fuchs equilibrium charge distribution. Subsequently, the aerosol enters a Differential Mobility Analyzer (DMA), where particles are classified based on electrical mobility. The classified particles then proceed to a Condensation Particle Counter (CPC, 377200), which determines their concentration. By scanning the voltage applied to the central rod, a particle number size distribution can be obtained, typically in the range of 10 to 800 nm. The MPSS is capable of detecting down to 10 particles per cm³ per scan (Wiedensohler et al., 2012a).




We processed the MPSS data following the protocol as outlined by Wiedensohler et al. (2012). The particle mobility was inverted to obtain particle number size distributions through bin width normalization and multiple charge correction. We accounted for particle losses by considering the equivalent lengths and individual flow rates of all components from the impactor outlet to the CPC inside the MPSS. Additionally, we corrected for internal losses caused by diffusion and ensured

accurate counting efficiency of the CPC. The MPSS participated in calibration workshops in July 2020 and February 2023 (World Calibration Center for Aerosol Physics, TROPOS). During our experiment, the measurements of the MPSS had an overall uncertainty of ±10%.

**DMS500**

The DMS500 (Cambustion) operates with a corona diffusion charger mounted on a classification column, which includes a central rod held at high voltage surrounded by a series of collection rings connected to sensitive electrometers along the length of the column (Symonds et al., 2007). During operation, charged particles enter the classification column and are carried towards the bottom by a flow of sheath air. The electric field from the central rod deflects particles towards the collection rings, where particles with higher charge-to-aerodynamic drag ratio are collected nearer the top of the column. The

electrometers detect the signals produced by the charged particles. Like this, the particle concentration can be determined simultaneously at the collection rings covering sizes from about 5 nm to 2.5 µm with a time-resolution of milliseconds. The DMS500 is designed for high particle concentrations, e.g. for applications in the field of engine development. Thus, the air sample is diluted twice before classification leading to a comparatively low sensitivity of 170 cm$^{-3}$ for 80 nm particles (Cambustion Ltd., 2019). Symonds et al. (2004), demonstrated that the DMS500's orifice, internal pressure, and aerosol

residence time had minimal impact on volatile aerosols. This is important for our study, as we tested the impactors also with semi-volatile secondary organic aerosol from alpha-pinene as precursor (see 2.2.2).

In our experimental setup, the 5-meter long heated line from the end of the impactor to the DMS500 was operated in an unheated state. The use of the line was necessary to accommodate the primary dilution and manage the pressure drops of the

impactors. To prevent any alteration in particle characteristics, we operated the line without heating. Before the measurements, we conducted a size calibration using polystyrene latex spheres with certified mean diameters (3320A/3495A, Thermo Scientific, Waltham, USA; NIST, National Institute of Standards and Technology traceable). Data were processed using DMS 6.09, the operational software, complemented by DMS Excel Utilities 7.49. To ensure comparability with CPC for smaller particles, the counting efficiency of the DMS500 is corrected by the software to 50% (and more) for particles with a diameter

of 23 nm (and less)(Solid Particle Measurements with a DMS500, 2024). The overall uncertainty of this instrument was in the range of ±23 %. This value was calculated as the relative standard deviation of the instrument's signal for the three test particle mixtures (see 2.2.2).



### 2.2.2 Testparticle generation

To determine the cut-off characteristics of the individual impactors under controlled conditions, we used a laboratory test-bed

consisting of a defined particle source, the test impactor, and the reference instrument to determine particle number size distributions, such as the MPSS or DMS500. If needed, the impactors were modified to have a cut-off at 100 nm of the last stage (see 2.1.). Furthermore, we removed the after-filter holder in order to directly assess the transmitted particles with the online reference instruments. Instead, after the impactor a y-piece made of stainless steel (inner diameter 15 mm) was installed directing the air flow to the detector and a pump. The pump was adjusted to achieve the correct overall sampling flow of the

impactor. The test impactor was then connected to one of three different particle sources. The set-up is presented in Figure S1 (Supporting Information).

**NaCl particles**

First, we generated salt particles from sodium chloride (NaCl). The particles were generated via a commercially available

nebulizer (PARI, LC SPRINT, Type 023). The nebulizer was filled with a saturated solution of NaCl in millipore water. We connected synthetic compressed air from the house line to the nebulizer and regulated it with a needle valve to 4 L/min. This aerosol flow was directed into a 30.5 L quartz glass flow tube for diluting, mixing, aging and drying. Since the majority of the impactors have a flow rate of 30 L/min, a dilution of 40 L/minsynthetic air was also fed into the flow tube. At the flow tube outlet, the test impactor was connected in line with the reference instrument and the pump. This set-up allowed us to run tests

for a time period of about 6-8 hours with a stable and relatively broad test particle number size distribution (Fig. 1).

With an estimate of density ($\rho_p$=2.16 g/cm³) and shape ($\chi$=1) of the NaCl particles, we converted the electrical mobility diameter ($d_m$) given by the reference instruments to aerodynamic diameter ($d_a$) (Baron and Willeke, 2011). The aerodynamic diameter is the basis for the size-based separation of particles in the impactors and thus the critical value for evaluation and

comparison.

$$d_a = d_m \cdot \sqrt{\frac{\chi \cdot \rho_p}{\rho_0}} \qquad (2)$$

**Soot particles**

Secondly, we tested the four impactors with diesel exhaust particles containing soot following the experimental setup as previously described by Mühlbauer et al. (2016). For our investigation, we employed a contemporary four-cylinder diesel engine supplied by Daimler AG (Model OM 651, Mercedes Benz, Germany). This engine is equipped with a common-rail





system and features direct-acting piezoelectric injectors from Delphi Automotive PLC (United Kingdom). In its default
configuration including a particle filter, this production engine complies with the European exhaust emission standard Euro 5.
The engine was run with a speed of 1200 rounds per minute (rpm), an injection pressure of 700 bar, a throttle position
maintained at 15%, and an exhaust gas recirculation (AGR) rate of 45%. The engine emissions were directly channelled from
the exhaust pipe to the impactor to achieve a stable particle number size distribution around the 100 nm diameter range (Fig. 1).

**SimSOA**

Thirdly, we tested the impactors' performances with atmospheric simulated secondary organic aerosol (SimSOA). In order to
conduct our tests at a stable mixture for several hours, we used one of the Bayreuth ATmospheric simulation CHambers
(BATCH) (Ofner et al., 2011). Before each measurement, the 700 L cylindrical glass chamber was flushed with outdoor-air to
introduce real environmental particles and trace gases. Subsequently, 0.05 mL of alpha-pinene was injected into the air flow,
while a nebulizer delivered seed particles (saturated ammonium sulfate solution) into the chamber at a flow rate of 3 LPM for
3 minutes. After an additional 5 minutes, the pump supplying outdoor-air was switched off, and the solar simulator (UV Osram
HMI, 4000 W, filtered with water-cooled glass plate, (Ofner et al., 2011; Zhao et al., 2008) was turned on for 15 minutes to
stimulate ozone production and the subsequent formation of SimSOA. Prior to commencing the impactor measurements, the
reference instruments were directly connected to the chamber to ensure suitable particle concentration and number size
distribution (Fig. 1).





**Figure 1: Particle number size distribution of three particle types for physical impactor testing - averaged for the duration of the experiments, and displayed with standard deviations for each particle type. The data for SimSOA (green) and NaCl (blue) are aligned on the left axis, while the soot data (black) is displayed on the right axis. SEM (Scanning Electron Microscopy) pictures of the test particles are displayed in Pictures A (SimSOA), B (NaCl), and C (Soot).**

### 2.2.3 Determining physical parameters as evaluation criteria

To comprehensively characterize the four cascade impactors for UFP sampling, we focussed on determining the pressure-drop, cut-off properties, evaluating potential particle losses, and identifying any measurement artefacts that may arise. For each impactor we followed the same protocol: (step 1) determine the particle number size distribution of the test particles, (step 2) insert the (modified) impactor and monitor the transmitted particles as described, (step 3) conduct an "empty" measurement after removing all impaction and nozzle plates from the impactor so that it included only the aerosol inlet and detector outlet.



The third step allowed for baseline measurements without the influence of impaction stages while maintaining a consistent flow rate to ensure integrity and comparability with the initial impaction measurements.

**Pressure-drop assessment**: The pressure drop across each impactor was measured using a differential pressure gauge. This measurement encompassed the total pressure loss through the impactor, including the quartz fiber after-filter that was integrated within the system. For this measurement, T-junctions were installed at both the inlet and outlet of the impactor. Branches from these junctions were connected to the differential pressure gauge (HONEYWELL, 26 PCCFA6D Differential Pressure Sensor, relative, $\pm$ 15 psi) to record the pressure drop across the entire impactor and the quartz fiber filter on which the UFP are collected. The pressure drop was determined under the same flow rate that was used for each impactor under operating conditions.

**Cut-off determination:** To derive the transmission of particles through the impactor as a function of their diameter, the ratio of the number concentration between the impactor (step 2) and the empty measurement (step 3) was calculated for each size bin. The resulting transmission curve was subsequently normalized for each impactor to the maximum transmission. From this, the dp50 values were deduced from a linear regression in the range of $30<d_m<110$ nm for NaCl and SimSOA particles and in the range of $60<d_m<110$ nm for soot. Additionally, we determined the steepness of the cut-off curve, and the dp90 and dp10 values, describing the diameter ranges at which 90% and 10% of particles, respectively, are secluded relative to the overall transmission of the impactor.

**Evaluation of losses:** We analyzed the original transmission curves for the ultrafine section of the particle size distribution for evaluating the losses of each impactor. Due to the relatively larger uncertainties in the reference instruments for very small diameters, i.e. $d_m <20$ nm, we decided to evaluate the particle number concentration at 30 nm for determining the losses in the ultrafine fraction. At this size all impactors showed a peak in the transmission curve which we used as a reference for determining the losses, calculated as 1 minus the derived transmission.

**Observation of other artefacts:** We also evaluated the transmission curve for other possible artefacts, as for example, for the transmission of larger particles, which potentially occurs as a result of particle bounce.

**2.3 Overview of field application of the four cascade impactors for chemical analysis of organic markers in UFP**

To assess the performance of the impactors under real environmental conditions, we operated all impactors simultaneously at the same location for the same duration for a side-by-side comparison. We took three samples of ambient aerosol on January 24th 2023, January 25th 2023, and January 30th 2023, in Bayreuth, Germany, for 24 hrs each. The impactors and their respective pumps were positioned indoors in the laboratories of the Bayreuth Center for Ecological and Environmental Research (BayCEER, 49.9305° N, 11.5881° E) situated in a semi industrial environment. Ambient air was drawn to each



impactor through antistatic inlet lines each 1.2 m long (3/8", TSI). The lines, impactors and pumps were maintained at a constant temperature of about 21°C. Each inlet line was equipped with honeycomb ceramic bodies (25,4 x 50mm; 400 CPSI, Rauschert, Germany) coated with Sodium Thiosulfate (Merck, ReagentPlus®, 99%) right in front of the impactors' inlet. The

coated ceramic body was hooded inside a stainless-steel housing and served as ozone denuder, preventing potential oxidation reactions on the already collected particles during sampling. The ozone scrubbers were previously evaluated for their capacity to scrub ozone from the sampling air and for potential losses ($\leq$ 6% for particles smaller than $d_m$=200 nm, $\leq$ 11% for particles larger than $d_m$=200 nm) (Eckenberger et al., in preparation). The flow rate was adjusted before each measurement for each impactor.


All upper stages of the impactors were coated with a thin layer of vacuum grease (Apiezon L, Apiezon products, M&I Materials Ltd, Manchester, England) to ensure the adherence of deposited particles and minimize bounce. Therefore, we first dissolved vacuum grease in n-hexane (Merck, ReagentPlus®, $\geq$99%) and applied approximately 0.05 mL of this solution onto aluminiumfoils using a syringe. We allowed the solvent to evaporate for a minimum of 12 hours before mounting the treated

foils onto all existing stages of the 120R MOUDI, the ultraMOUDI, and the ELPI. For the PENS, which does not have an impaction plate suitable for filter installation, the grease was applied directly to the plate following the evaporation process. After each 24-hour sampling period, all impactor and nozzle plates were cleaned using an ethanol-water mixture, then dried with synthetic compressed air, and recoated before the next measurement. For collection of UFP, we inserted pre-baked quartz fiber filters (Whatman QM-H, 47mm or 37mm) in the after-filter holder of each impactor. These were stored at -20°C

immediately after collection.

For comparing the results of chemical analysis of the sampled UFP between the four impactors, we selected six organic marker components to encompass a mixture of anthropogenic (Benzo[a]pyrene (BaP), Benzo[b]flouranthene (Bbf)) (Hussain et al., 2018) and biogenic sources (Pinic acid (PA), Terpenylic acid (TA)(Vestenius et al., 2014)). Furthermore, we analyzed

Levoglucosan (Levo) as marker for biomass burning (Simoneit et al., 1999) and N-(1,3-dimethylbutyl)-N'-phenyl-p-phenylenediamine (6PPD) as marker for tire wear material (Klöckner et al., 2021). The marker components were selected to provide a diverse range of molar masses, volatility, and mass abundance.

### 2.3.1 Extraction of samples

We extracted the selected marker components from the filters via a soft, solvent-based and optimized protocol: (1) the filter

loaded with particles was divided into two equal parts. One part was extracted, the other one used as backup. (2) The filter-half for extraction, was spiked with 50 µL of each internal standard, namely 3-Methylcholanthrene (3-MC, 0.4 µM) and Nicotinic acid (NA, 10 µM) and cut into small fragments. (3) These filter fragments were then transferred into a glass container with a screw cap, and 2 mL of extraction solvent (e.g. analytical-grade Dichloromethane (DCM, Fisher Chemical, 99.8%) and



Methanol (MeOH, Carl Roth, ≥ 99.9%)) were introduced. (4) The samples underwent extraction through agitation within a
closed flask for a duration of 15 minutes using a vortex shaker (2000 rpm). (5) Filter residues were kept in the glass container. Extracts were filtered using specially designed glass frits with a diameter of 1 cm and a pore size of 20 µm to eliminate any potential filter residue.

Steps (3) to (5) were repeated three times, each time employing a different extraction solvent. The sequential solvents used
were, in order, pure MeOH, 50:50 MeOH:DCM, and pure DCM. Subsequently, the solvent from the combined extracts was evaporated under a gentle flow of Nitrogen ($N_2$, 99.99%) while cooled with ice to avoid loss of semi-volatile compounds. A droplet was kept as residue which was dissolved in 1 mL of a 60:40 solution of Acetonitrile (ACN, Carl Roth, 99.95%) and Millipore water ($H_2O$, obtained from Seralpur PRO 90 CN system with Supor DCF filter, Electronics Grade, 0.2 µm). This was transferred into a separate vial for subsequent analysis. Throughout the entire sample preparation process, the samples
were stored in an ice cooled environment.

### 2.3.2 HPLC analysis for chemical marker profiling

The analysis of the sample extracts was performed at two High Performance Liquid Chromatography (HPLC) systems. For the detection of PA, TA, Levo, and 6PPD, an Agilent series 1100 chromatograph equipped with an electrospray ionization
mass spectrometer (ESI-MS, Agilent 6130 single quadrupole) was utilized. To determine the concentration of the polycyclic aromatic hydrocarbons (PAH), BaP and Bbf, an Agilent 1260 Infinity system coupled with a fluorescence detector (FLD, Agilent 1100 Series) was employed. For the mobile phase, HPLC-grade acetonitrile, Milli-pore water, and formic acid as buffer (HCOOH, Carl Roth, p.a. ≥ 98%) was used. We applied three different methods which are summarized in Table S1. For each analyte, uncertainty propagation was based on the relative standard deviation of repeated analysis (2%), errors from
reference material (0.5%), errors from self-prepared stock solutions (5%), dilution errors (5%), calibration uncertainty (5%) and inaccuracies from the balance and pipette (1%). The overall relative uncertainty for each analyte was calculated by combining these individual uncertainties, resulting in an overall relative uncertainty for each analyte of approximately 9%.

To assess the efficacy of the sample preparation, including the extraction method, recovery experiments were conducted in
triplicates. Half of the pre-heated quartz fiber filter (Whatman QM-H, 47 mm) was spiked with 10 µL of a 10 µM standard solution encompassing all markers. Subsequently, the spiked filter was extracted as outlined previously. The recoveries (Rec) were ascertained utilizing external standard (ESTD) calibration and calculated by dividing the measured concentration of each marker by the expected (spiked) concentration.

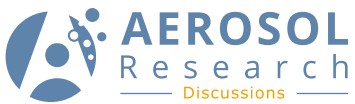


$$\text{Rec} = \frac{c_{measured}}{c_{expected}} \cdot 100 \tag{3}$$

The obtained recoveries were as follows: BaP 78±7%, BbF 74±7%, Levo 79±7%, PA 84±8%, TA 85±8%, and 6PPD 75±7%. As these recovery rates describe the average systematic loss of the marker compounds during extraction, we corrected our data

using the following equation:

$$c_{sample_{corrected}} = c_{sample_{measured}} \times \left( \frac{100}{\text{Rec}} \right) \tag{4}$$

The recovery-corrected results were then compared against the NIST standard and showed agreement within the uncertainty of the measurement, even in the presence of a particulate matrix. The limit of detection (LOD) was determined based on the

standard deviation (σ) of the response from a modified calibration solution, targeting a signal-to-noise ratio of approximately 3 for each analyte, as well as the slope/response factor (RF) from the ESTD calibration.

$$\text{LOD} = \frac{3 \cdot \sigma}{\text{RF}} \tag{5}$$

The LOD was calculated across four replicates. Furthermore, the LOD for airborne concentrations (LOD$_{Air}$) [pg/m$^3$] was calculated by dividing the LOD by the sampling volume. For the MOUDI and ELPI devices sampling for 24h with 30 L/min, the sampling volume was 43.2 m³. For the PENS, which has a sampling flow of 4 L/min, the sampling volume was 5.76 m³. The LOD values ranged around 1-2 and 40-186 pg/m³ for target markers measured with HPLC-FLD and HPLC-MS, respectively. The LOD increased by a factor of 7.5, when sampling a smaller volume as for the PENS compared to all other

impactors (Table S2, Supporting information). Additionally, we estimated impactor-specific overall measurement uncertainties using Gaussian error propagation, with uncertainties ranging from 13.8% to 17.8% (SI 1.1).

## 3. Results and Discussion

### 3.1 Physical characterization I: pressure drop, cut-offs, steepness of cut-off curve

The **pressure drop** across an impactor including the quartz fiber filter can significantly influence the collection efficiency and the chemical composition of the captured particles. The more the pressure drops across the impactor and quartz fiber filter, the





higher the rate of potential evaporation of semi-volatile compounds. For the four tested impactors, the pressure-drop was 260±1 (PENS), 420±2 (ultraMOUDI), 600±3 (120R-MOUDI), and 690±3 (ELPI) hPa.

Figure 2 shows the normalized transmission curves obtained from testing the four impactors with the three test particle mixtures. **Cut-offs** were calculated as dp50 for the electromobility diameter. They increased consistently for all impactors from NaCl particles over SimSOA to soot particles (59-68 < 70-74 < 102-116 nm). The best agreement between the individual impactors' cut-offs was found for SimSOA. For SimSOA, the ultraMOUDI and 120R MOUDI had the smallest cut-offs with an electromobility diameter of 71±7 and 70±7 nm. Contrastingly, for soot, the largest deviations among the impactors' cut-off

were found, showing that the devices deviated about 14 nm at most. Here, the two MOUDI-models had highest cut-offs and the PENS the smallest one with 102±10 nm. For NaCl particles, we compared the aerodynamic cut-off diameters, which were 86±9 (ultraMOUDI), 89±9 (120R MOUDI), 99±10 (PENS), and 100±10 (ELPI) nm.

Generally, the transmission curves were steeper for soot than for SimSOA than for NaCl particles. This trend is strongest for

ELPI and PENS, which have the highest **steepness** for soot with slopes of the transmission curve of -1.1 1/nm and the least sharp cut-off for NaCl with -0.52 and -0.56 1/nm, respectively. The sharpness of cut-off of the 120R MOUDI and ultraMOUDI seems less sensitive to the type of test particle, as the steepness of the transmission curves varied about 9% at most between the different test particles. The four impactors had best agreement regarding the steepness of the transmission curve for SimSOA with a relative difference between maximum and minimum slopes of about 26%.


The **dp10 and dp90** represent the diameters at which 10% or 90% of the test particles were secluded by the impactor. An impactor with closely spaced dp10 and dp90 values would have a steeper transmission curve, indicating a more precise segregation of particles by size. For all impactors, the lowest dp10 values were reached with NaCl particles (dp10ave=34 nm, average across all impactors), indicating a relatively lower efficiency for capturing small NaCl particles in comparison to

SimSOA (dp10ave=39 nm) and soot (dp10ave=74 nm). The dp90 did not vary systematically between the test particles and impactors ranging from 125±13 nm (for NaCl particles collected with ultraMOUDI) to 183±18 nm (for SimSOA collected with 120R MOUDI). As this general trend might be driven by the relatively stickier nature of NaCl particles leading to a relatively smeared cut-off, it is interesting to observe the sharpest cut-off for the soot particles, which exhibit the most complex shape of the three tested particle types. This can be seen also from the difference between dp90 and dp10, which was smallest

for soot particles for all four impactors.

SimSOA is likely the most representative test particle for environments ranging from urban to suburban to rural which are not coastal, kerb site or subject to nearby combustion sources. For the tests with SimSOA, the four impactors performed comparably in terms of cut-off, transmission curve steepness, and dp10 and dp90 (Table 1). We observed a tendency for a

sharper separation of UFP with PENS and ELPI compared to the two MOUDI-models. However, relatively larger deviations





between the four impactors was observed for soot particles, both in cut-off and steepness. We derived the cut-off as aerodynamic diameter only for NaCl-particles and found that both, ELPI and PENS have a UFP suitable cut-off within the uncertainty. The two MOUDI-models underestimate the target UFP cut-off of 100 nm slightly. Yet, the cut-off determined for NaCl-particles is at the lower end when compared to the other test particles. Thus, with more realistic air mixtures, we consider that the ultraMOUDI and 120R MOUDI would shift in their cut-off to slightly larger diameters and therefore the two MOUDI-models can be suitable for collecting UFP as well.



**Table 1: Summary of performance test results for the four impactors and the three test aerosol mixtures. The dp values are provided as electromobility diameter in nm. The steepness describes the slope of the transmission curve around particles diameters of 100 nm. The dp50$_a$ (aerodynamic particle diameter at 50% collection efficiency) is calculated as aerodynamic cut-off diameters derived for the salt particle tests.**

| Impactor | Aerodynamic cut off [nm] (manufacturer) | Aerosol type | dp value [nm] | | | Steepness [1/nm] | dp50$_a$ [nm] |
|---|---|---|---|---|---|---|---|
| | | | 10 | 50 | 90 | | |
| 120R MOUDI | 100 | NaCl | 34±4 | 61±6 | 182±18 | -0.55 | 89±9 |
| | | SimSOA | 41±4 | 71±7 | 183±18 | -0.55 | / |
| | | Soot | 78±8 | 116±12 | 175±18 | -0.86 | / |
| PENS | 100 | NaCl | 30±3 | 67±7 | 177±18 | -0.52 | 99±10 |
| | | SimSOA | 42±4 | 74±7 | 148±15 | -0.64 | / |
| | | Soot | 69±7 | 102±10 | 153±15 | -1.1 | / |
| ELPI | 90 | NaCl | 39±4 | 68±7 | 162±16 | -0.56 | 100±10 |
| | | SimSOA | 39±4 | 71±7 | 148±15 | -0.65 | / |
| | | Soot | 72±7 | 106±10 | 162±16 | -1.1 | / |
| ultra MODUI | 100 | NaCl | 32±3 | 59±6 | 125±13 | -0.67 | 86±9 |
| | | SimSOA | 35±4 | 70±7 | 162±16 | -0.58 | / |
| | | Soot | 76±7 | 112±11 | 166±17 | -0.95 | / |






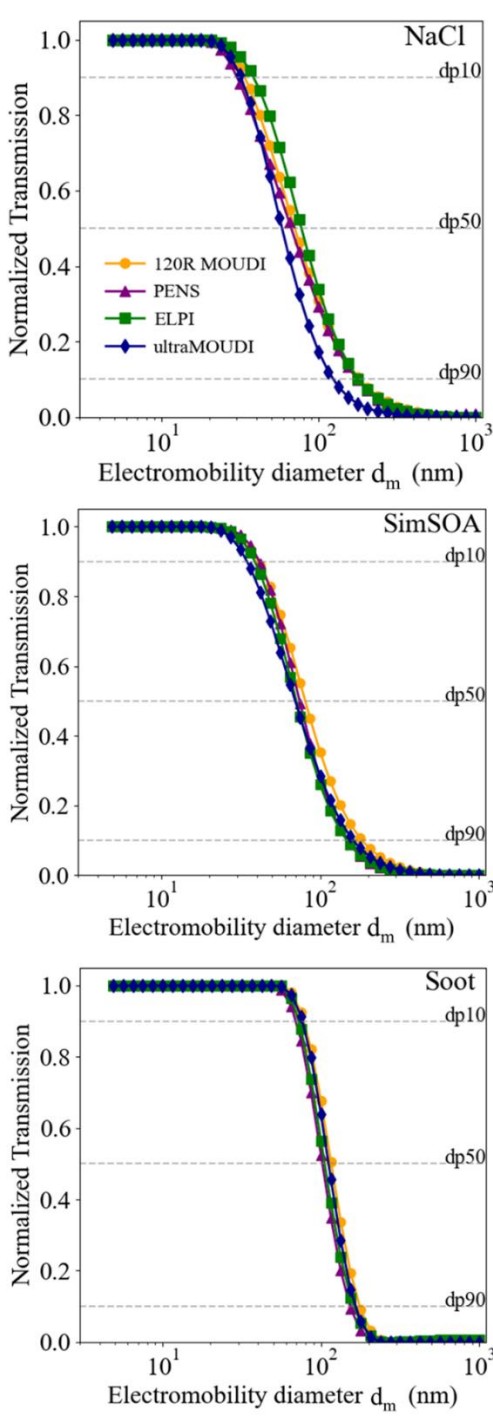

**Figure 2: Normalized transmission curves across three test particle types including NaCl, SimSOA, and soot, as captured by different impactors: 120R MOUDI (orange, circle), PENS (purple, triangle), ELPI (green, square), and ultraMOUDI (blue, diamond markers). Each subplot displays the transmission versus electromobility diameter (nm) on a logarithmic scale. Lines are included to guide the eye.**




## 3.2 Physical characterization II: losses and particle bounce

For determining potential **losses** within the impactors, we compared their original transmission curves for SimSOA generated in the atmospheric simulation chamber (Fig. 3). The maximum transmission in the ultrafine fraction at 30 nm was 84±8% for

PENS, 77±8% for ultraMOUDI, 75±8% for 120R MOUDI, and 69±7% for ELPI. These results indicate that the ELPI had the highest losses of SimOA UFP during sampling compared to the other tested impactors. However, considering the overlap of uncertainties, these differences are not significant. Losses can be due to wall losses or interstage losses, retaining particles on surfaces other than the impaction plate. In addition, losses can also occur through evaporation of (semi) volatile particles, especially under high pressure drop during flow-through (Marple and Willeke, 1976; Won Kim, 2010).


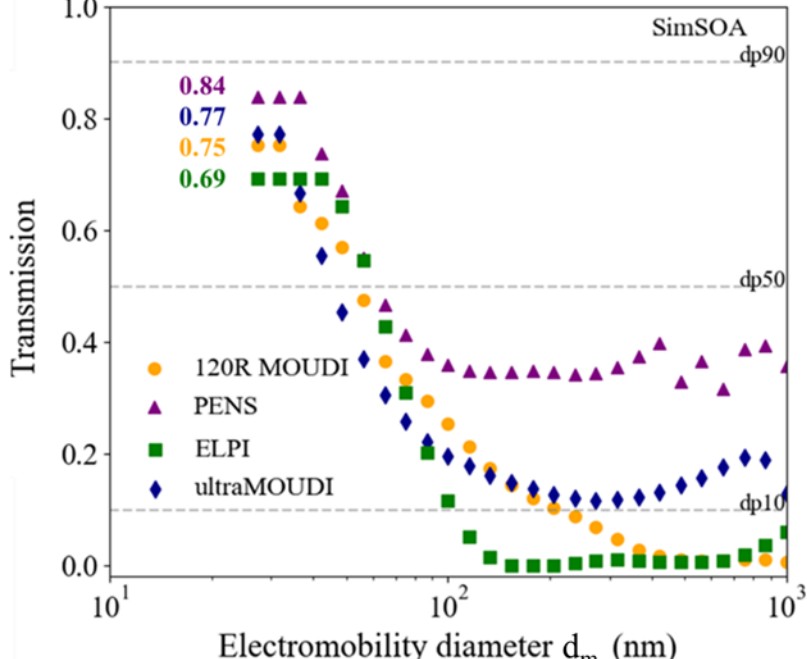

**Figure 3: Transmission curves for the four tested impactors collecting SimSOA generated within the atmospheric simulation chamber. Note that all tests were performed with the original impactors and no coating was applied. The four impactors are marked**

**as 120R MOUDI (orange, circle), PENS (purple, triangle), ELPI (green, square), and ultraMOUDI (blue, diamond markers).**

Ideally, the transmission should ultimately reach zero for particles with larger diameters, as these should be retained in the upper stages of the impactor. However, the elevation for diameters between 200 and 1000 nm above zero suggests the occurrence of **particle bounce** (Fig. 3). The ELPI reached a transmission of about 0-6% for larger particles. Thus, within the

uncertainty, no particles with an electromobility diameter larger than 177 nm passed the last impaction stage. The same can be



observed for the 120R MOUDI, yet, it reached the point of no transmission at a relatively higher electromobility diameter of 562 nm. For larger particles, the transmission remained zero. This is in line with the comparably flat slope of the transmission curve of the 120R MOUDI, indicating a broad separation of particle sizes rather than a sharp cut-off (Table 1). However, both PENS and ultraMOUDI had notable bounce effects. For the PENS, the transmission dropped to a local minimum at 133 nm
(electromobility diameter) and remained at approximately 38% for larger particles. For the ultraMOUDI, the local minimum of transmission was at an electromobility diameter of 273 nm with 13%, but then increased again for larger particle diameters to about 19%.

Overall, the impactor design seems to critically impact potential losses and particle bounce. The ELPI and the 120R MOUDI
have a higher number of stages, impaction and nozzle plate pairs, before ultimately separating UFP (13 and 9 stages, respectively), when compared to ultraMOUDI and PENS (3 and 2 stages, respectively). The increased number of stages appears to reduce the bounce effect (Fig. 3). This is likely because particles have more opportunities to impact as they traverse through multiple stages, and each stage is less loaded with particles, reducing the likelihood of bounce. However, this comes at the cost of comparably increased losses in the UFP range. This contrasts with the ultraMOUDI and the PENS, where fewer
stages or a distinct design approach, like the cyclone pre-separator in PENS, result in fewer losses. It has been reported, that particle bounce becomes particularly significant for lower cut-off stages because of the gradual reduction in pressure at each stage, which subsequently leads to a decrease in relative humidity (RH). The reduction in RH in turn can intensify the particle bounce effect (Chen et al., 2011). Pressure dropped most drastically throughout the ELPI, which could be problematic for collecting semi-volatile organic marker compounds (Yao et al., 2022). Knowing that the diameters of cut-off and the effective
sharpness of the separation between fine and ultrafine SimSOA particles were comparable for all tested impactors, the impact of losses and particle bounce on a mass based chemical analysis might be significant, which therefore is further investigated in the following.

Considering the mass of UFP compared to the mass of potential artefacts, we wanted to understand the relative impact of
particle bounce and losses. Even a few larger particles or fragments bouncing onto the UFP collection substrate could heavily skew the results, leading to an overestimation of the UFP mass concentration and a misrepresentation of their chemical composition. This is particularly problematic because UFP can have different chemical properties and health impacts compared to larger particles (Abdillah and Wang, 2023; Müller et al., 2012). Particle bounce-off can be influenced by the impaction surface, the presence, type and depth of a coating, particle types, particle loading, sampling conditions, and the impaction
substrate. Particularly during long-term sampling with heavy particle loads, deposited particles were found in excessive layers (Kulkarni et al., 2011; Marple et al., 1991; Pak et al., 1992; Turner and Hering, 1987; Chang et al., 1999; Newton et al., 1990;Lai et al., 2008; Rao and Whitby, 1978).



To test whether increased particle load leads to more severe particle bounce, we varied the particle number concentration in the chamber for SimSOA from a maximum of about 200,000 cm$^{-3}$ to a more realistic maximum of about 5,700 cm$^{-3}$ (see Figure 4). As a wide range of number concentrations was covered, we measured the particle number size distributions with both, DMS500 (Level 1, in Fig. 4a) and MPSS (Level 2, 3, and 4, in Fig. 4a). Additionally, Figure 4a presents an exemplary number size distribution (Level 4) for ambient particles which was collected during the period of environmental sampling when the impactors were deployed side-by-side (see 2.3. For these levels, Figure 4b shows the transmission through the PENS impactor as an example. Indeed, reducing the particle number concentration resulted in a decreased bounce effect, although it remained noticeable. Compared to the highest particle load (Level 1), transmission was reduced in Level 2 for particles sized 150-200 nm, while particles > 500 nm still exhibited transmission rates of up to 34%. Level 3a was the relatively lowest particle load, which then decreased transmission to about 5-9% remaining relatively consistent across the diameter range of 170-800 nm.

Furthermore, greasing of upper stages is a method that has previously been suggested and applied to reduce particle bounce (Baron and Willeke, 2011; Ungeheuer et al., 2021). To test this method, we greased the upper stages of the PENS (Fig. 4 right panel, Level 3b). The application of the coating on the impactor further reduced the bounce effect, lowering transmission rates to 0-6% within the same particle size range. The duration of these measurements is notably brief (20 minutes), in contrast to the typical collection times employed with impactors (e.g. 24 hr). Thus, we greased also the upper stages of the 120R MOUDI and monitored the transmission on three consecutive days while sampling ambient air. While the greasing improved the sharpness of the cut-off curve, the transmitted fraction of particles around a size of about 200 nm increased during the three days from about 0 to 15%. These tests highlight the variations in the observed bounce effects over short and extended collection periods, which may be very much dependent on particle load and nature. Yet, greasing the upper impaction stages overall improved the UFP sampling as it reduced the fraction of relatively larger particles being transmitted.





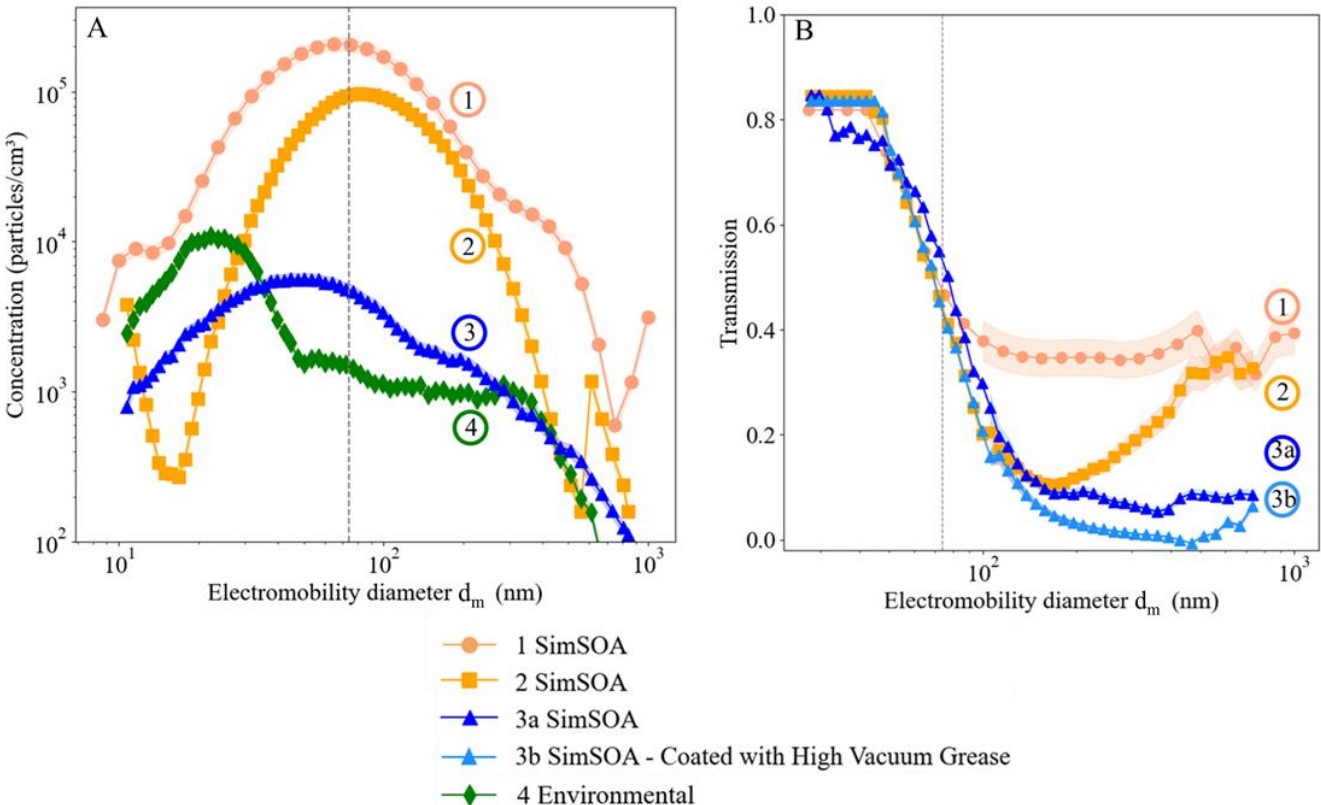

**Figure 4: A illustrates the particle number size distribution in units of particles/cm³ for SimSOA (1, as detected in highest concentrations with DMS500, 2 and 3, in relatively lower concentrations measured by MPSS), and environmental SOA (4). B delves into the transmission curves of the PENS for the respective particle size distribution. Data labeled with 3a and 3b display uncoated and coated measurments using high vacuum grease.**

### 3.3 Physical characterization III: transmitted particle mass

Typically, UFP collection using impactors is followed by the analysis of the chemical composition which have a mass based focus. Breakthroughs of coarser particles into the UFP range can significantly distort mass based analytical results, as particle mass increases in cubic proportion to the diameter. As an example, Figure 5 illustrates the transmission of particles through the PENS based on their respective mass. The density of the alpha-pinene particles was measured by Zelenyuk et al. (Zelenyuk et. al., 2008) as 1.20 g/cm³. We used this value to calculate the mass of particles transmitting the impactor according to the measured number size distributions assuming spherical shapes of the SimSOA particles. We separated the data into two size classes (based on the particle's electromobility diameter): (1) the entire recorded range 10-800 nm and (2) the UFP range 10-74 nm which is below the previously determined dp50 (section 3.1).



The set of measurements includes particle mass as measured directly from the chamber, from the empty PENS, the uncoated PENS, and the coated PENS (Fig. 5). For particles ranging from 10 to 800 nm, the mass transmission, calculated directly from the particle number size distribution of the chamber, was 220.2 µg/m³. The mass of the transmitting particles through the empty

PENS was 205.87 µg/m³, through the uncoated PENS it was 21.4 µg/m³, and through the coated PENS 3.3 µg/m³. This highlights the coated impactor's capability to capture over 98.5% of the test particle mass, whereas the uncoated impactor captured approximately 90.3%. The transmitted ultrafine fraction remained comparable between coated and uncoated tests.

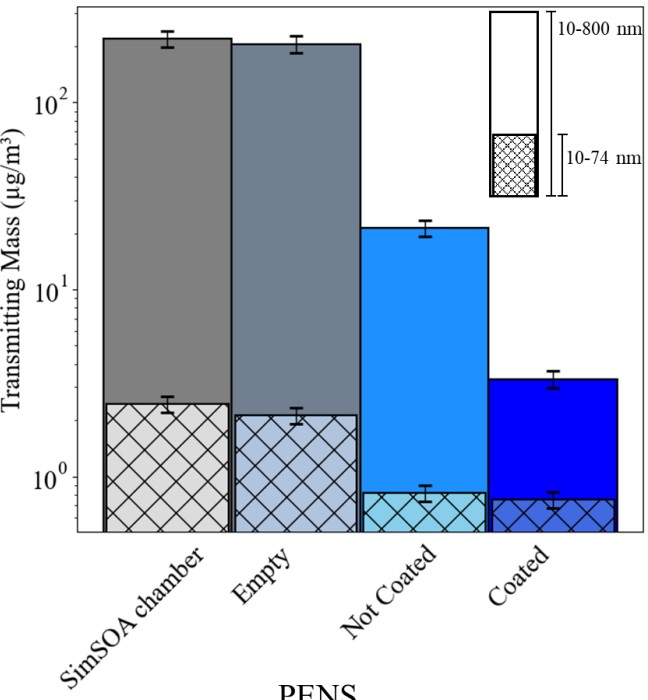

**Figure 5: Comparative bar chart displaying the mass of transmitting particle mass in µg/m³ for coated and not coated PENS tests compared with the original aerosol mixture (SimSOA) and the empty PENS. Two segments present the entire observed electromobility diameter range (10...800 nm) and ultrafine fraction (10...74 nm). The chart employs a logarithmic scale on the y-axis.**




**3.4 Field application: chemical analysis of organic markers**

We evaluated the performance of the four cascade impactors under environmental conditions. We selected six markers, which represent typical UFP sources, but also cover a wide range of molecular properties (e.g. molecular weight, vapour pressure, functionality).

- Polycyclic aromatic hydrocarbons (PAH) have potential impact on human and environmental health (Hussain et
al., 2018; Kim et al., 2013; Wang et al., 2006). **Benzo[a]pyrene** (**BaP**, M= 252.31 g/mol) and **Benzo[b]fluoranthene** (**BbF**, M= 252.31 g/mol) are high molecular weight PAHs with five aromatic rings, low volatility and predominantly found in fine particulate matter originating from incomplete combustion.

- **Levoglucosan** (**Levo**, M = 162.14 g/mol) is a well-studied tracer for cellulose combustion (Bhattarai et al., 2019; Simoneit et al., 1999). Its presence is indicative of residential wood burning, agricultural fire practices, and
wildfire emissions. Levoglucosan is semi-volatile and can partition between gas- and particle-phase (Xie et al. ,2014).

- **Pinic acid** (**PA**, M = 186.21 g/mol) and **terpenylic acid** (**TA**, M = 198.24 g/mol) are representative for SOA of biogenic origin. These organic acids are produced through the oxidation of terpenes, such as alpha-pinene and beta-pinene, emitted by vegetation. PA and TA have a semi-volatile nature and thus partition between gas- and
particle-phase (Claeys et al., 2013; Grieshop et al., 2007).

- **N-(1,3-dimethylbutyl)-N'-phenyl-p-phenylenediamine** (**6PPD,** M=268.40 g/mol) is an additive of tire wear material and has been recently proposed as tracer for non-exhaust traffic-related particles. 6PPD is thought to be stable in the particle phase, however, it can react with atmospheric oxidants to form oxygenated products (Chen et al., 2023; Hu et al., 2022).


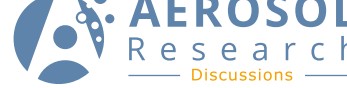

**Figure 6: A : Bar chart showing the average mass concentration of selected organic marker compounds in UFP collected with PENS (purple), ELPI (green), ultraMOUDI (dark blue), and 120R MOUDI (orange) of three days in ng/m³ on a logarithmic scale. Error bars indicate standard deviations. B: Relative deviation (%) in mass concentration of BaP, BbF, Levo, PA, TA, and 6PPD in UFP from the average of all impactors including and excluding PENS for the caluculation of the average (transparent/opaque bars). For comparison, horizontal grey lines represent the estimated overall error for each marker and impactor (SI 1.1). A logarithmic representation of 6b can be found in SI Fig 2.**





We analyzed UFP sampled in an urban, semi-industrial environment. The observed average mass concentrations are presented in Figure 6A. Furthermore, we evaluated the relative over- or underestimation of each impactor compared to the average of all impactor results (Figure 6B). Overall, the best agreement between all impactors was found for the two PAHs. For BaP, the average mass concentration was highest for the PENS with 227.2 ± 97 pg/m³ (average ± standard deviation over the three consecutive days), which is 29% above the average of 175.65 pg/m³ when considering all impactors. ELPI and ultraMOUDI

agreed with about 4% above and 3% below the average, respectively. The 120R MOUDI had the lowest average concentration with 122.6 ± 74 pg/m³, which was 30% below the overall average. For BbF, the tendencies were the same but less pronounced. We can highlight two findings here for the analysis of the two PAHs: Firstly, the maximum disagreement can be as large as 37% between PENS and 120R MOUDI. Secondly, the two PAHs, originating from the same sources and found in >90% particle phase, exhibit the same sampling tendencies across the impactor models. This is likely due to their identical molecular

weights and very comparable boiling points.

As the results of the PENS seemed to be consistently on the higher end for all marker compounds, we calculated the average between all impactors including and excluding the PENS and then compared (Figure 6B). The PENS exceeded the average of the remaining impactors by a factor of 1.3 for BaP, 1.2 for BbF, 2.0 for Levo, 1.5 for PA, 1.7 for TA, and 2.3 for 6PPD. Our

cut-off tests indicate three potential reasons for this overestimation. Firstly, the PENS had the lowest pressure drop of all impactors (260 hPa), which might affect the gas-particle partitioning of semi-volatile compounds in a way that e.g., Levo, TA, and PA had comparably higher mass concentrations in PENS samples. Secondly, the bounce effect was more pronounced in the PENS and dependent on the particle number concentration of the SimSOA. Possibly and despite the applied coating, the overestimation in marker mass concentrations of the PENS could be due to a bounce effect, which is increasing with rising

ambient mass concentrations. Thirdly, the physical loss in the UFP range was relatively small in the PENS when compared to the other impactors (Figure 3). The losses, however, were similar in all impactors and likely less impactful from a mass-based viewpoint.

The comparison of the remaining impactors showed the best agreement for the results of 6PPD. Herein, the two MOUDI

models had on average 83±30 pg/m³ and the ELPI 86±48 pg/m³ of 6PPD in the sampled UFP. The differences amongst the 120R MOUDI, ELPI, and ultraMOUDI increased from 6PPD (-2 to +3%), to PAHs, BaP and BbF, (+8 to -22%), the organic acids, PA and TA, (+20 to -35%), and, Levo (+51 to -31%). Thus, the range of deviation increased with the average observed mass concentration from 6PPD (84±30 pg/m³), to BaP (158±25 pg/m³) and BbF (199±23 pg/m³), to PA (3,144±620 pg/m³) and TA (2,231±541 pg/m³), and finally to Levo (40,660±17,148 pg/m³).


It should be noted that the differences among the impactors fall within the same order of magnitude as the overall uncertainty for most markers (BaP, BbF, TA, and PA). Yet, these differences were systematic and seem to not only depend on the impactor design but also on the properties of the analyzed compounds (see 2.3.2).





**3.5 Influence of physical factors on the results of the marker mass concentration analysis in UFP**

We analyzed systematically the influence of the previously examined physical factors on the results of the marker concentration in UFP with respect to the markers properties. While the cut-off diameter, cut-off curve steepness and losses seem to not vary largely between the four tested impactors, the pressure drop and the effect of particle bounce have the potential to drive the observed differences of the mass based UFP analysis.

We expected the pressure drop to primarily affect the semi-volatile markers such as PA, TA, and Levo, with their mass concentrations likely being reduced in an impactor with a higher pressure drop due to evaporation during sampling. Figure 7a illustrates the dependency of the absolute deviation of the average mass concentration excluding the PENS (as a reference point) on the pressure drop for all samples and impactors. The absolute deviation of the average mass concentrations of PA, TA, and Levo decreased with increasing pressure drop, from 2.3 to 0.46 ng/m³/hPa for PA (PENS to 120R MOUDI), 2.8 to 685 0.62 ng/m³/hPa for TA (PENS to ELPI), and 80 to 7.6 ng/m³/hPa for Levo (PENS to 120R MOUDI). The less volatile markers, BaP and BbF, also showed a decrease in mass concentration with increasing pressure drop, although less pronounced.. As previously noted, the mass concentration of 6PPD as determined with the PENS (260 hPa) was significantly elevated compared to all other impactors. For ultraMOUDI, ELPI, and 120R MOUDI, which ranged in pressure drop from 420 to 690 hPa, the mass concentration of 6PPD was stable. These findings seem to confirm that the larger pressure drop in the impactors leads to 690 evaporation and thus a mass loss of the semi-volatile substances. Yet, the trends of BaP, Bbf and 6PPD indicate additional influencing factors beyond the pressure drop alone.

Despite the application of a coating, we suspected particle bounce and thus tested whether the absolute deviation from the impactor average mass concentration was dependent on the ambient marker mass concentration (Fig. 7b). In case of particle 695 bounce, we expected an increasing effect with increasing mass concentration. Indeed, this increase can be observed for all impactors throughout the entire range of observed marker mass concentrations. Comparing the extend of deviation of the impactor average as a function of mass concentration and by the help of a linear regression curve, we noticed that the PENS had the largest slope (2.05), followed by the ultraMOUDI (0.46), and ELPI (0.28) and 120R MOUDI (0.18). These slopes represent the gain of falsely UFP-attributed mass per ambient marker mass, likely caused by particle-bounce, and compare 700 well to the results of the laboratory tests (Fig. 3).

While the nature of the particle could play a role in its efficiency of being captured in the filter substrate - such as comparing sticky, spherical SOA to combustion particles with complex shapes (Boskovic et al., 2005; Matthew et al., 2008; Huang et al., 705 2004)- the design of the impactors likely determines the potential for particle bounce. The main difference between the PENS and the other impactors is the cyclone and the number of stages used for separating and impacting larger particles. The larger number of stages and, thus, the larger number of coated surfaces in the two MOUDI models and the ELPI are likely beneficial



for reducing the impact of particle bounce. The rotating stages of the 120R MOUDI potentially played a more significant role in the field study with longer sampling times than in laboratory tests. Over time, the rotation resulted in an even loading of the

upper stages, which in addition to the coating likely reduced particle bounce. Thus, we observed quasi no dependency of the 120R MOUDI deviations from the average mass concentration with increasing mass concentration (Fig. 7b). However, as the marker compounds with a high observed mass concentration are at the same time semi-volatile, we cannot completely separate the two effects of pressure drop driven losses and particle bounce effects within this study.

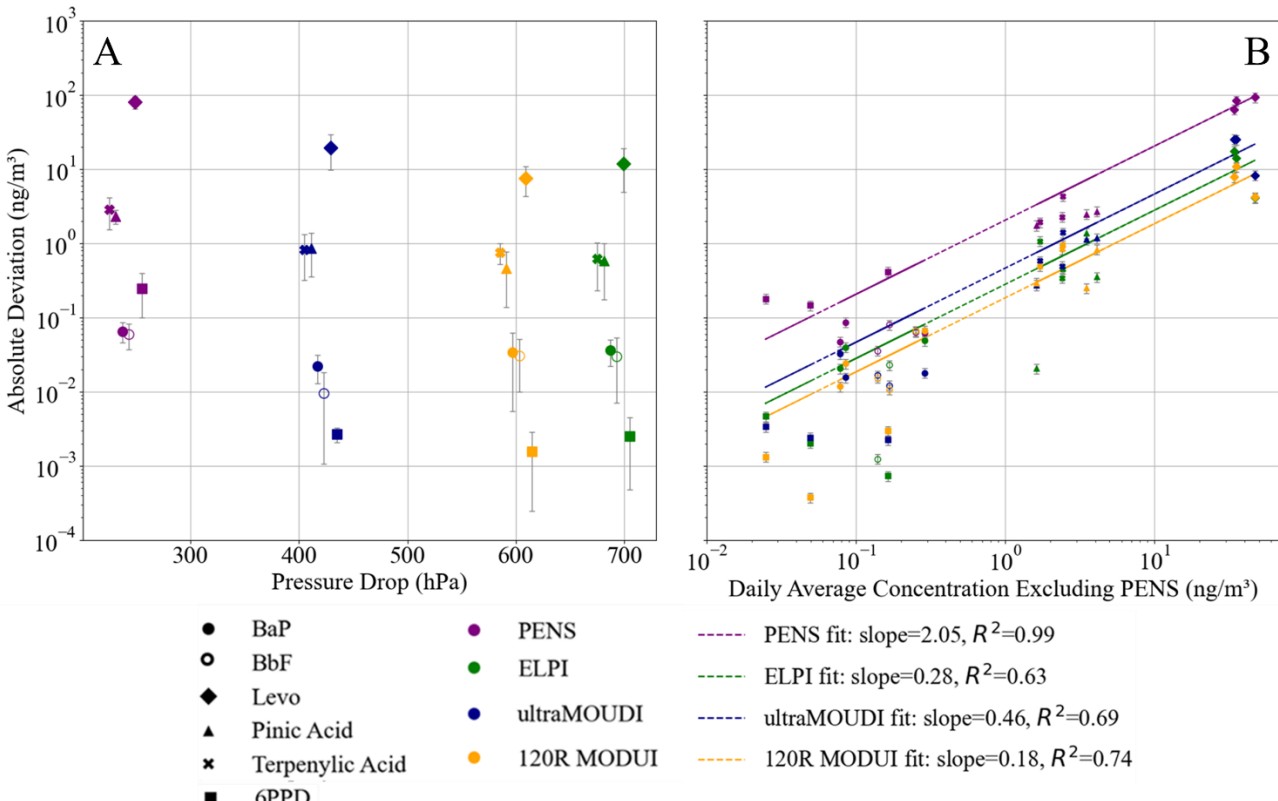

**Figure 7: (A) Absolute deviation of average mass concentrations of BaP, BbF, Levo, Pinic Acid, Terpenylic Acid, and 6PPD as determined with the four impactors as a function of pressure drop across the impactors. Data points are displaced horizontally around each impactor's nominal pressure drop value for reasons of illustration to allow comparison while avoiding overlap. Markers denote different substances: BaP (filled circle), BbF (unfilled circle), Levo (diamond), Pinic Acid (triangle), Terpenylic Acid (X), and 6PPD (square). (B) Absolute deviation from the mass concentration average for PENS (purple), ELPI (green), ultraMOUDI (dark**

**blue), and 120R MODUI (orange) for the selected organic marker compounds in UFP. Linear regressions through the origin are included for each impactor, with slopes and R² values indicated. The legend presents substances, impactors, and linear regression fits.**




## 4. Conclusion

This study provides a detailed characterization and comparison of the performance of four impactors 120R MOUDI, ultraMOUDI, ELPI, and PENS. To understand the impact of the impactors' design on its ability to collect UFP, we examined their pressure drop, cut-off diameters, steepness of cut-off curve, losses, and particle bounce. Under controlled conditions using three test particle mixtures, we showed that not only the impactor design but also the nature of the particles is a reason for differences in the observed results. This was confirmed when we applied the four impactors to ambient air to examine the mass concentrations of six organic markers in UFP. Summarizing, all four impactors were capable of separating and collecting UFP as they all have a cut-off diameter of about 100 nm. More explicitly, they can be characterized as follows:

- The PENS is compact and lightweight, thus portable and suitable for field or personal exposure studies that require mobility. With a sampling flow rate of 4 L/min a smaller air volume is probed when compared to other impactors. This affects the detection limits, while keeping the pressure drop within the device low (about 260 hPa). Thus, the evaporation and mass-loss of semi-volatile compounds is comparably small. The PENS showed an impact of particle bounce, particularly at higher particle concentrations. We showed that the application of a coating eliminated the transmission of larger particles in the laboratory tests. However, we observed a high deviation from the other impactors results in the field comparison when markers were apparent in high mass concentrations. This is likely due to the design of the PENS which has a cyclon for pre-separation and only one stage for impaction of larger particles that could be used for applying a coating. Yet, for the particle-bound BaP and BbF in moderate mass concentration, the PENS compared well with the other impactors.

- The ELPI was simplified for this study. When operated in its original set-up, it has the advantage of a parallel online monitoring of the number size distribution. Here, the ELPI showed a sharp cut-off for UFP and effectively no transmission of larger particles. Likely this is due to its design with 13 nozzle and impaction pairs, which can retain larger particles and prevent fragments to cascade through the device. However, this comes with the cost of a relatively high overall loss in the ultrafine fraction (69±7%) and a high pressure drop of 690 hPa. The reduced pressure on the filter substrate can lead to the evaporation of semi-volatile substances during sampling. This limits the ELPI's suitability for analyzing these compounds, but for stable, particle-bound compounds in UFP (e.g. metals) it can be very applicable.

- The 120R MOUDI has a rotating design, that should distribute particles evenly across the impaction surface, reducing particle build-up and re-entrainment. Indeed, no transmission of larger particles than about 500 nm was observed in our test and the bounce-effect remained comparably small. However, the cut-off curve was relatively broad as can be seen for example from the dp90, which was with about 180 nm the highest among the four tested impactors. Interestingly, in our tests the sharpness of the cut-off curve for the 120R MOUDI was not impacted by the particle





type. The 120R MOUDI has a moderate pressure drop of 600 hPa, thus caution is required for semi-volatile substances. For future studies, it would be interesting to determine the long-term efficacy of the rotation in minimizing the bounce effect.

- The ultraMOUDI is a non-rotating and reduced variant of the 120R MOUDI. With a moderate pressure drop of 420
hPa, it is less likely to cause evaporation of semi-volatile compounds compared to the ELPI and 120R MOUDI, but it still maintains a relatively sharp cut-off curve. Due to a reduced number of nozzle-impactor pairs, it showed fewer losses in the ultrafine range compared to the ELPI and 120R MOUDI. When compared to the ELPI and 120R MOUDI on the basis of the mass concentrations in ambient UFP, the ultraMOUDI agreed well within the measurement uncertainty for the mostly particle-bound BaP, BbF, and 6PPD, and found slightly higher concentrations for the semi-
volatile TA, PA, and Levo. Similar to the PENS, we observed transmission of larger particles due to potential particle bounce, yet showed that this effect was reduced with the application of a coating. The ultraMOUDI is smaller and simpler in the handling than the 120R MOUDI and thus can be integrated in automated, standalone low-volume samplers, for instance.

Our findings indicate that separating and collecting UFP for mass-based chemical analysis is challenging. Numerous factors affect the separation and collection of UFPs, complicating the comparability of studies. The nuanced performance differences among the impactors underscore the need for careful consideration of the intended application and the potential artefacts that may arise during sampling and analysis. Each impactor offers unique advantages and limitations, making it essential to match the impactor to the specific research goals and the properties of the particles being studied. Furthermore, the variability between
the impactors' performance for the analysis of the six selected organic markers suggests that factors such as chemical composition, particle morphology, and physical interactions with the impactor significantly influence the results, as well.

**Acknowledgement**

We gratefully acknowledge the financial support of this project by the Bavarian State Ministry of the Environment and
Consumer Protection. We also thank Wolf-Ulrich Palm and Klaus Kümmerer from University Lüneburg for providing us with the fluorescence detector. We extend our gratitude to the Bavarian Polymer Institute (BPI) Key Lab at the Bayreuth University for the scanning electron microscopy images. We are grateful for useful discussions with Sarmite Kernchen.

**Code and data availability**

The code and data used in this study are available by request to the corresponding authors.

**Competing interests**

The authors declare that they have no conflict of interest



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
