# Peer review of "Performance evaluation of four cascade impactors for airborne UFP collection: Influence of particle type, concentration, mass and chemical nature"

_Aerosol Research, 2024_

## Author Comment (AC1)

This review provided valuable feedback regarding our manuscript "Performance evaluation of four cascade impactors for airborne UFP collection: Influence of particle type, concentration, mass and chemical nature". We are grateful for this insight, appreciate the opportunity to clarify a few points within this discussion, and will address the concerns raised point-by-point in the following text (Comment & Answers). Furthermore, we used the reviewers advice to improve the manuscript and highlight those modifications subsequently (Modifications).

**Major comment 1:**

The authors selected 2 commercial impactors that have since long been used for ambient sampling (MOUDI, ELPI) and two more compact instruments that do not seem to be commercially available (ultraMOUDI, PENS). Unfortunately, however, the commercial and established impactors were used in a rather unconventional way, by removing one or more of the lower stages and replacing them with a quartz-fibre after-filter. These modifications likely change the performance characteristics of these impactors compared to their original configuration and the results of the present study are therefore difficult to transfer to other applications of MOUDI and ELPI. Quartz filters have different sampling characteristics as the typically applied flat impaction substrates (in terms of adsorptive and evaporative and bounce-off artifacts, for example), the pressure drop is different in the modified version and possibly also the internal air flow characteristics have changed. What was the motivation to apply these modifications instead of using the impactors as intended by their manufacturers? In any case, the modifications should be made very clear already in the abstract and potential impacts on the UFP sampling performance should be discussed in the manuscript.

Answer 1:

This major comment is valuable as it shows that the description of the specifications of the tested impactors and their modifications is missing information:

Firstly, we would like to clarify that the ultraMOUDI and PENS impactors, although not commercially available off the shelf, are indeed obtainable from the manufacturers upon request. These instruments were designed for our specific purpose of UFP collection and separation. Specifically, we asked to include the 100 nm stage as a last separation stage. Additionally, to the established ELPI and MOUDI instruments, we also wanted to test a more compact version to offer a space-efficient alternative, potentially integrating these impactors into a mobile module equipped with an automated filter changer.

https://dekati.com/products/elpi/
https://tsi.com/products/cascade-impactors/moudi-ii-rotating-impactors/moudi-ii-impactors-120r,-122r,-125r/
https://digitel-ag.com/product/low-volume-aerosol-sampler-dpa-14/

Secondly, it seems there has been a misunderstanding regarding the modifications we applied to the MOUDI and ELPI impactors in the study. Thus, we revised the manuscript to improve

clarity. You raised the concern that we used the MOUDI and ELPI in a rather unconventional way, which indeed would be critical for the overall conclusions of this study. Yet, all we did is typically done by most researchers who intend to sample organic markers in the ultrafine fraction using these two impactors. To do so, we omitted the lower stages without substituting them to achieve a cut-off diameter at 100 nm. The after-filter holder that we employed is the manufacturer-supplied after-filter stage, standard practice when using these instruments. For reference, we several studies have used QFF with impactors when interested in similar organic target analytes, such as Marple et al., 2014; Mazzi et al., 2024; Fujitani et al., 2006a; Kim et al., 2002; Wang et al., 2024.

Thirdly, we agree with your observation that the use of quartz-fiber filters (QFF) might result in changes to the flow and sampling dynamics due to their structure. However, as outlined in the manuscript, the upper stages of the MOUDI, ELPI, and ultraMOUDI were installed with the manufacturer-supplied aluminum filters, thus maintaining the intended sampling characteristics in those stages.

Moreover, that the QFF after-filters were used exclusively during the field study. In contrast, for the transmission measurements conducted under laboratory conditions, no after-filters were applied in order to allow for accurate transmission measurements.

We will make sure to address these points more clearly in the abstract, methods, and throughout the manuscript, as well as discuss the potential impacts of these modifications on UFP sampling performance in greater depth.

**Major comment 2:**

In some places in the manuscript, the authors seem to imply that UFP sampling is only "correct" with a cut-off diameter exactly at 100 nm (e.g. L125 or L481-485). Given that the value of 100 nm is rather a convention than a physical law, I would suggest to relax such statements. Properties and impacts of particles change rather gradually than abruptly at 100 nm, so slight deviations from a nominal 100 nm cut-point are likely not very relevant, especially with regards to mass-based sampling and analyses as done in this study.

Answer 2:

We discussed the "usefullness" of exact 100 nm cut-offs extensively during the preparation of this manuscript. We acknowledge that the 100 nm cut-off is more of a convention than a strict physical boundary. Additionally, we could discuss to what extend a strict cut-off at 100, 200, 300, 400 or 500 nm would change our view about nature and impact of airborne sub-μm particles (Kittelson et al., 2022). However, we felt this goes beyond the scope of our manuscript and aimed for addressing the emerging need of chemically well characterized ultrafine particles. Finally, we agree that particle properties change gradually, and slight deviations of ±10 nm around the nominal 100 nm cut-point are unlikely to significantly affect our mass-based analyses. Therefore, we noted in the text that all the impactors used in the study showed comparable cut-off characteristics. As suggested by the reviewer, we checked the text to further relax our statements regarding the cut-off

Modification 2:

Please find the amendments e.g. in Modification 52.

**Major comment 3:**

More important are potential artefacts from evaporation and bounce-off. These are discussed in several places of the manuscript, but often in rather qualitative and speculative ways only. I give some suggestions below,but would like to encourage the authors to try and use their data to improve the quantitative understanding of these critical issues in UFP sampling.

Answer 3:

This feedback is a valuable observation to us. One of our main results from this study is indeed that the artifacts from evaporation and bounce-off are more critical than the cut-off itself.We greatly appreciate your specific suggestions, and we will incorporate them into the manuscript, taking care to address these artifacts in a more quantitative manner, where possible, later in the text.

Modifications 3:

Please, see specific comments e.g. No. 6, 12, 18, 58, 63

**Major comment 4:**

The Introduction suffers from lengthy parts describing textbook knowledge or irrelevant historical developments, while lacking a proper appreciation of the literature of ) impactor comparisons, including for nanoparticle characterisation, and b) applications of impactors for chemical UFP studies.

Answer 4:

Ok, we adjusted the introduction accordingly by shortening sections that contain textbook knowledge or historical developments. Additionally, we ensured to include a more comprehensive review of the literature, particularly focusing on impactor comparisons for nanoparticle characterization and the application of impactors in chemical ultrafine particle studies.

Modifications 4: for example

L79-90: Removed paragraph about physical background

L95-96: Removed sentence

L101-111: Removed historical background

L111-116: Adopted text to account for previous changes

**Major comment 5:**

In many places, the manuscript is written in slightly imprecise language, which is inappropriate for a scientific manuscript. Some examples follow below, but more can be found throughout the manuscript and a careful check by the more experienced co-authors should be done to eliminate any ambiguity and impreciseness.

Answer 5:

We carefully reviewed the manuscript to address any imprecise language and ambiguity, making the necessary revisions to ensure clarity and precision throughout.

Modifications 5:

Please, find these modifications directly in the revised manuscript.

**Specific comment 6:**

The abstract could be improved by shortening the more descriptive parts and including more quantitative results, especially for the field intercomparisons.

L15-16 "techniques developed for larger particles": Unclear, which techniques are referred to here. Impactors have since long been applied for sampling in the UFP size range as well.

L22-24: Unclear sentence structure, please rephrase

L24 "successfully": Did the authors expect otherwise? Is it really worth noting in the abstract that the impactors work as designed and intended?

L29: evaporative and bounce-off artifacts have long been known as critical issues in impactor sampling. It might be misleading to argue they were identified in this study.

L31: Based on the results that follow later, I would argue they were partially mitigated at best.

L35: What does "semi-industrial air" mean?

L40: …were about 29% higher of 30% lower…?

L45: …with decreasing pressure drop…?

L46: Isn't it trivial that marker concentrations increase with higher air concentrations? What do you mean here?

Answer 6:

Thank you for the specific suggestions. We have revised the abstract accordingly.

Modifications 6:

L15-16: "However, their small mass challengessampling and chemical characterization methods."

L22-24: "We observed that the performance of the impactors varied between 59 and 116 nm in cut-off diameter. This was depending on the impactor design and the type of test aerosol mixture, which was either salt particles (NaCl), simulated secondary organic aerosol (SimSOA), or soot. All impactors separated UFP..."L29: "We compared two additional impactor-specific factors crucial for mass-based analysis of organic marker compounds: evaporation of semi-volatile compounds due to high pressure drop across the impactor and material addition from larger particles bouncing off upper stages.was influenced by particle number concentration in the sampled air and could be partially mitigated by applying a coating to the upper impaction plates.L3: "In the field application, we deployed the four cascade impactors side-by-side under environmental conditions to sample urban air."

L39-41:"BaP had an average mass concentration of 175±25 pg/m³ across all impactors and sampling days, but concentrations  about 29% higher when sampled with PENS or 30% lower when sampled with 120R MOUDI, indicating a maximum disagreement of nearly 60%."

L45-end of the abstract were rewritten completely to comprise the overall message further as:

"The mass concentrations of the semi-volatile markers, PA, TA and Levo, were decreasing on average from PENS, to ultraMOUDI, to 120R and ELPI. We attributed this tendency to two effects: (1) Likely evaporation losses of those markers were driven by the pressure drop within the impactor, which was indeed increasing from PENS to ELPI. (2) Despite the applied coating, bounce-off might have affected the smallest impactors (PENS and ultraMOUDI) the most as these have less stages for retaining larger particles and fragments."

**Specific comment 7:**

L38: The inclusion of 6-PPD as a typical UFP marker is interesting. Is there evidence that tyre wear is a primary source of UFP? Given the mechanical abrasion process, I would expect these particles in the coarse fraction only.

Answer 7:

Löber et al., 2024 found that ranged different particle sizes. Specifically, ultrafine particles (UFP) were observed in the 10 nm range, particularly during high-speed driving and acceleration phases. Additionally, a broader size distribution of particles between 10 nm and 100 nm was identified, with a secondary mode between 60 and 90 nm, particularly during acceleration events. Larger particles, ranging from 300 nm to 10 µm, were also detected, particularly in association with road dust resuspension and thermal processes. This size distribution highlights that tire wear can contribute to both UFP and larger particle fractions. We decided to include 6PPD here, also because of its characteristics e.g. a particle-bound nature.

**Specific comment: 8:**

L94: These might not be the best references for the statement, as they do not seem to deal with UFP composition.

Answer 8:

We have added an additional reference that specifically addresses the extraction of UFP particles from impactors.

Modifications 8:

"Particles caught in the substrate can be extracted and undergo chemical analysis to determine their composition (Bein and Wexler, 2014; Canepari et al., 2010; Daher et al., 2011)"

**Specific comment 9:**

L124-129: Misleading and confusing paragraph, please rephrase. The cut-off diameter is a property of any impactor stage, it does not "determine a threshold" to distinguish fine from ultrafine particles. The aerodynamic diameter is just one of several diameter definitions, it is not "an abstract measure describing a sphere…". And the cut-off diameter does not have a "sharpness", which varies with particle properties.

Answer 9:

As the paragraph presents our expectations and thus is important for the discussion of our results, we appreciate the direct and constructive feedback. We have removed the definitions and have instead listed only (1) what determines the cut-off diameter in an impactor and (2) how real atmospheric irregular particles might affect the separation characteristics.

Modifications 9:

"The **cut-off diameter** for UFP (dp50=100nm, referring to aerodynamic diameter) depends on diameter, nozzle size, flow rate, distances between plates, and its shape. , real atmospheric particles will vary in impaction behaviour and thus influence ."

**Specific comment 10:**

L131-132: What do you mean by "larger" and "smaller" particles here? I doubt sedimentation plays a role inside an impactor.

Answer 10:

We have removed the sentence.

**Specific comment 11:**

L133-134: Particles can bounce without breaking apart as well! What does "falsely adding material" mean?

Answer 11:

We have removed "falsely" and clarified the sentence accordingly.

Modifications 11:

"During collection either larger particles bounce or break into fragments and re-entrain adding mass to the filter substrate dedicated for UFP or target UFP bounce-off from the collection substrate and are lost for analysis."

**Specific comment 12:**

L136-138: Such evaporative losses are likely different between quartz-fibre filters, where particles are exposed more individually and directly to ventilated air as compared to compact deposited material on flat impaction substrates.

Answer 12:

Yes, likely the evaporative losses depend on the collection substrate.  we tested various substrates before starting our study and found that the QFF were a good trade-off for sampling and extraction procedures. As we wanted to raise this aspect here in the introduction, we feel like the general statement still holds (independent from the filter used). Yet, we integrated this comment in the discussion part of our manuscript.

Modifications 12:

L690-L691: „ Likely, the evaporative loss depends also on the collection substrate. When sampled in QFF, for example, UFP are exposed individually and ventilated efficiently. However, here QFF was used in all impactors for UFP sampling and. Furthermore, we firstly noted differences between 120R MOUDI and ELPI, despite of a comparable pressure drop. Secondly, mass concentrations of 6PPD were comparable for all impactors except for the PENS. These observations indicate additional influencing factors beyond the pressure drop alone, which could potentially affect the analysis more significantly."

**Specific comment 13:**

L138: How would evaporative losses lead to mass gain? What do you mean by "hampered conclusion", conclusion on what?

Answer 13:

It is possible that condensable gases adhere to the surface of the collected particles on the surface. In this case we would expect a gain of mass from condensation of gaseous semi-volatile compounds. Yet, we see from the reviewers comment that the statement was misleading and rephased the sentence.

Modifications 13:

"Semi-volatile particle-bound components may **evaporate** from already collected particulate surfaces during the ongoing sampling due to a reduced pressure within the impactor and continuous ventilation. This can result in alterations of the chemical composition due to interactions with the gaseous phase."

**Specific Comment 14:**

L148: Why have exactly these 4 impactors been selected? Have other models been considered?

Answer 14:

We considered several impactors for the separation and collection of atmospheric UFP when we prepared this study. Our selection criteria were as follows: The device needed to be suitable for continuous operation. The impactor needed to collect all particles below 100 nm on a single after-filter, which ideally could be integrated in an automatic filter changer in future applications. Furthermore, we wanted to test devices with different designs, flow rates, and stage numbers. As outlined in the paper, we found that we were limited when the impactors exhibited too high pressure drops, due to our test setup and with respect to expected losses regarding semi-volatile marker compounds.

For these reasons, we focused on the four selected impactors: ELPI and 120R MOUDI, which have been used in several previous studies, the PENS, which differs in design and has a comparably small flow rate, and a compact version of a MOUDI. However, we are aware that several other cascade impactors exist and that new models are recently or currently developed.We made this clear in the text now.

Modifications 14:

L 148-151: "Today, several cascade impactors exist, which are either commercially available or newly developed (Crazzolara and Held, 2024; Ngagine et al., 2022;Järvinen et al., 2014; Romay and García-Ruiz, 2023;Marple et al., 2014a; Tsai et al., 2012b)"

**Specific comment 15:**

L150: I guess the pressure was regulated between impactor and pump, but the air flow rate was measured at the impactor inlet? The current phrasing might be slightly misleading.

Answer 15:

Thank you, we have clarified this point in the text.

Modifications 15:

L150 "The flow rate was determined at the impactors inlet with a Gilibrator 2 bubble flow meter and re-checked following each collection interval."

**Specific comment 16:**

L158: While the MOUDI does use 47mm substrates, the area the particles are deposited on is actually smaller than 47 mm.

Answer 16:

Thank you, as this information is irrelevant for the reader at this point, we chose to remove the 47mm substrates from the sentence here. In line L167-168 we explicitly describe which substrates were used.

Modifications 16:

"It is further supported by rotating the impaction plates relative to the nozzles while distributing the sampled particles over the sampling substrates."

**Special comment 17:**

L162: What do you mean by "clean flow path"? How is it different (cleaner?) than with the MOUDI I version?

Answer 17:

The difference is the design of the rotating impaction plates. While the older models use external gears and hooks which reach through the sample flow, the new 120R-MOUDI II has integrated miniature stepper motors that rotate the impaction plate. This way no other obstacles than the nozzle and impaction plates are within the air stream. We used the wording from the literature: "In the 120 MOUDI-II, the flow path is considered "cleaner" from the impaction plate to the nozzle plate of the next stage compared to the original MOUDI" (Marple et al., (2014); Fig1). Nevertheless, we suggest rewording to "undisturbed".

Modification 17:

L162 "Therefore, the 120R MOUDI-II offers an undisturbed flow path from the impaction plate to the next stage's nozzle plate."

**Specific comment 18:**

L165 and 195: If the lowest stages of the MOUDI and ELPI are removed and replace with an after-filter holder, how does this change pressure drops and air flow rates through the impactors? Could this lead to different sampling characteristics of the upper stages and/or influence particle bounce?

Answer 18:

The MOUDI is a modular cascade impactor and the cut-off diameters in the stages can be selected by the user depending on the study's requirements. Thus, we did neither run the tests with missing stages nor with an additional after-filter holder. We chose the stages as outlined in the manuscript to be able to sample UFP as bulk without further separation. For the ELPI, this was not as straight-forward as we had to work with placeholders in order to remove the stages below 100 nm but maintain the outer dimensions of the device and spacing between the stages. Finally, we worked with the original impactor design, holder, after-filter, and pump.

We adjusted the flow rates for all cascade impactors according to the recommendations and then determined the pressure drop throughout the entire device. For the PENS, coarse particles are separated with a cyclon first and then a MOUDI nozzle plate is used. For the ELPI, the jet orifices are designed to decrease pressure efficiently early within the device.

It is possible that the sampling characteristics within the impactor potentially affecting particle bounce could vary for other models and stage selections. Our results show that the high number of coated stages is beneficial for reducing particle bounce, which could be observed for both MOUDI 120R and ELPI. This is indeed an interesting point raised by the reviewer, which we emphazised now stronger than previously in the paper's discussion part.

**Specific comment 19:**

L167-168: The given cut-points sum up to 8 stages only, while nine stages are mentioned in the text.

Answer 19:

In the MOUDI, each stage contains an impaction plate and a nozzle plate. A schematic of this can be found in Chien et al., (2015); Marple et al., (2014) for example.

[Figure]

**Figure 1: Chien et al., (2015); schematic representation of a nozzle impaction plate pair.**

1. Nozzle plate     5. Substrate holder
2. Nozzle plate holder     6. Upper casing for nozzle plate
3. Impaction substrate     7. Teflon gasket
4. Substrate clamp     8. Bottom casing for impaction plate assembly

The given list in our methods sections correspond to the manufacturer's specifications of cut-off diameters described as the dp50 of particles . For the MOUDI 120R-II, this means that we could collect ultrafine particles with dp50≤100 nm downstream of the lowest plate. The nozzle plate with 0.1 µm cut-off and the collection substrate in the afterfilter holder is here considered as first stage. If we would have been interested in size distributions, we could have collected particulate matter with 100nm<dp50≤180nm on the filter substrate on the second stage between the lowest and the next nozzle plate, and so on. Therefore, as described, we account for a total of 9 stages with 9 cut-off diameters (0.10, 0.18, 0.32, 0.56, 1.0, 1.8, 3.2, 5.6, and 10 µm). To avoid confusion, we carefully revised the wording in the methods section.

Modifications 19:

L153-156: "(Marple et al., 1991)."

L166-167: "It separated airborne particulate matter into nine fractions having nominal cut-off diameters (dp50) of 0.10, 0.18, 0.32, 0.56, 1.0, 1.8, 3.2, 5.6 and 10 µm."

**Specific comment 20:**

L168+175+199+209 and others: Were the applied after-filter holders custom-made or original parts of all four impactor models?

Answer 20:

The after-filter holders used were original parts provided with all four impactor models, not custom-made.

Modifications 20:

see Modification 14

**Specific comment 21:**

L171: Who developed the ultraMOUDI, the authors or MSP/TSI? Was it indeed developed specifically for this specific study?

Answer 21:

MSP/TSI offer a reduced version of the MOUDI, which has 3 non-rotating stages and a 30 LPM sampling flow. The comparably high flow rate (contrasting e.g. the PENS with 4 LPM) was interesting to us as we expected to be able to collect comparably more mass in the ultrafine fraction. We asked MSP/TSI to equip this MOUDI (e.g. model number 100S4) with different impactor and nozzle plate pairs than foreseen in the original device, namely with cut-off diameters at 0.1, 1 and 2.5µm.

To make this point , we included the model number and reworded the text slightly.

Modifications 21:

"We asked the manufacturer for three stages with cut-off diameters at 0.1, 1 and 2.5 µm."

**Specific comment 22:**

L173: The given cut-points sum up to 2 stages (0.1 – 1 and 1 - 2.5 µm), while three stages are mentioned in the text.

Answer 22:

As outlined in Answer 19, the UFP-stage (<0.1µm) is counted as stage itself. In case of the ultraMOUDI, UFP-stage, stage 0.1-1µm and stage 1-2.5µm add up to a total of three stages.

**Specific comment 23:**

L188: Same here, the number of given cut-points does not fit the number of stages

Answer 23:

Please view our argumentation of the response to comment 14 and 22. We, however, tried to improve the text to be more consistent.

Modifications 23:

 L194: "For the collection of UFP, we removed the stages with cut-off diameters of 0.03 and 0.06 µm to achieve a final cut-off size of 0.09 µm at stage 3."

**Specific comment 24:**

L207: How was the original PENS modified and by whom?

Answer 24:

We requested modifications from the manufacturer of the PENS, Haze Control System Inc., based on our specific needs for this study. Compared to the original design described in the first PENS publication (Tsai et al. 2012), we asked for a slightly higher flow rate 4 LPM instead of 2 LPM and a first separation within the cyclon at 2.5µm instead of 4µm. As the current formulation of our text seems to be misleading, we revised the paragraph accordingly.

Modifications 24:

L206-207: "The manufacturer modified the PENS for this study to separate the particles in the fractions 2.5 and 0.1 µm and to operate at a sampling flow rate of 4 L/min."

**Specific comment 25:**

L212 and elsewhere: I'd suggest to always use "cut-off diameter" or "cut-off size" or maybe "cut-points" instead of just "cut-off".

Answer 25:

We have made this adjustment at the relevant points throughout the text.

**Specific comment 26:**

Section 2.2.1: The detailed descriptions of the operating principles of MPSS and DMS are not really needed and could be replaced by proper references.

Answer 26:

Thank you for pointing out lenghty paragraphs in our manuscript. We have shortened the instrument descriptions significantly.

Modifications 26:

L223-237: " The MPSS (TROPOS) classifies and quantifies airborne particles based on the principle of the mobility of charged particles in an electric field (Wiedensohler et al., 2012a, 2018). Sampled particles are neutralized to achieve a Fuchs equilibrium charge distribution, then classified by a Differential Mobility Analyzer (DMA) based on electrical mobility. The particle number size distribution is obtained by scanning the voltage applied to the DMA, typically covering a range of 10 to 800 nm. The MPSS is capable of detecting down to 10 particles per cm³ per scan (Wiedensohler et al., 2012a). The MPSS data were processed

following the protocol by Wiedensohler et al. (2012), including corrections for particle losses and CPC counting efficiency. The MPSS measurements had an uncertainty of ±10%.

L240-262: "The DMS500 (Cambustion) uses a corona diffusion charger and a classification column with electrometers to detect particle concentrations across a size range of 5 nm to 2.5 µm with a time-resolution of milliseconds (Symonds et al., 2007). The air sample is diluted twice before classification, leading to a comparatively low sensitivity of 170 cm$^{-3}$ for 80 nm particles. The DMS500 was calibrated using polystyrene latex spheres (3320A/3495A, Thermo Scientific, NIST), and data were processed using DMS software (DMS 6.09 complemented by DMW Excel Utilities 7.49). The overall uncertainty of the instrument was ±23%, calculated based on the relative standard deviation for the test particle mixtures (Cambustion Ltd., 2019, Symonds et al. (2004), Solid Particle Measurements with a DMS500, 2024) (see 2.2.2).

**Specific comment 27:**

L253-255: Confusing. Why exactly was a 5 m long line needed in the first place? The line was not heated, as it seems, but in L253 it says "heated line". Which one is correct?

Answer 27:

The term "heated line" refers to the manufacturer's product name, which we used in the manuscript for consistency. However, we operated the line in an unheated state during the experiment to avoid artefacts. Due to the recent revisions and shortening of the section, this description was omitted.

Modifications 27

see Modifications 26

**Specific comment 28**

L270: Figure S1 is helpful in understanding the experimental setup and should be placed in the main manuscript.

Answer 28:

Ok, we inserted Figure S1 as Figure 1 into the revised manuscript.

**Specific comment 29:**

L285: Eq. 2 is textbook knowledge and not needed here. If so, it needs to be referred to and all variables explained.

Answer 29:

We removed Equation 2 as suggested.

**Specific comment 30:**

L295: Was the particle filter included or removed for this application?

Answer 30:

The particle filter was included in the engine configuration for this application, and the engine complies with the Euro 5 emission standard. We changed the text accordingly.

Modifications 30:

L295: "This production engine, equipped with a particle filter, complies with the European exhaust emission standard Euro 5."

**Specific comment 31:**

L303: Flushing the chamber with ambient air seems unusual. What are ambient particles and gases needed for in the chamber runs? Is the resulting SimSOA actually a mixture of chamber SOA and ambient aerosol? If so, what fraction of ambient aerosol particles does SimSOA contain?

Answer 31:

We intentionally flushed the chamber with ambient air to closely simulate real environmental conditions and create a mixture of chamber-generated SOA and ambient aerosol. This approach allowed us to perform our experiment with a similar complex, but less variable aerosol mixture as compared to real ambient air which we found important for testing the impactors in real-world atmospheric scenarios. This is why we first introduced ambient particles and gases into the chamber, then added a-pinene and seed particles, and finally irradiated the mixture. We found that without the a-pinene addition particle number concentrations were not large enough for the transmission tests conducted with the DMS500. Adding a-pinene and seed particles increased the particle number concentrations about a factor of .

**Specific comment 32:**

L323: I can't follow how step 3 was actually performed, please explain more. What is left, if all impactor stages are removed? Or do you mean the entire impactors were removed and the inlet line directly detected to the outlet line to pump and MPSS?

Answer 32:

The "empty" measurement in steps 3 refers to measurements that were conducted when the line up to the impactor inlet and the inlet itself remained the same as in step 2. Yet, all nozzle stages and impaction plates that make up the body of the impactor were then removed, so no size classification occured. At the end of the impactor, the corresponding outlet is connected, just as in step 2, with an additional line to the measurement device, which also remains consistent with step 2. We revised the text for more clarity as follows:

Modifications 32:

L322-325: "... (step 3) conduct an "empty" measurement. For the "empty" measurements all nozzle stages and impaction plates were removed, so no size classification occurs, and inlet and outlet of the impactor were connected just as in step 2. Step 3 allowed for baseline measurements without the influence of impaction stages while maintaining a consistent flow rate to ensure comparability with the transmission measurements of step 2."

**Specific comment 33:**

L325: The motivation for step 3 is not entirely clear to me. Did these measurements show different concentrations than step 1, i.e. PNSD upstream the impactor? If so, why?

Answer 33:

Indeed, we wanted to ensure that possible effects from the lines or the impactor inlet and outlet would be the same as for the transmission measurement in step 2. Thus, we conducted both measurements with conditions that were as similar as possible. Typically, we detected differences in the range of 8-12%.

Modifications 33:

See Modification 32

**Specific comment 34:**

L339, Figs. 2&3: Check your definitions and explanations of dp10 and dp90. Do you mean dp90 refers to the diameter at which 90% of particles are deposited at the pre-UFP stage, i.e. 10% of particles at this diameter are collected as UFP?  for dp10, the phrasing is unclear. The diameters are defined in different ways in Figure 2 and 3, please harmonize.

Answer 34:

Thank you for the careful review and for bringing this inconsistency to our attention. We have clarified this section in the text and adjusted Figure 3 as follows: We revised the definition of dp10 and dp90 which should refer to the particle diameters at which 10% or 90% of the particles are deposited and hence retained in the impactor  not transmitted.

Modification 34:

L339-L341: "Additionally, we determined the steepness of the cut-off curve, and the dp90 and dp10 values, describing the diameters at which 90% and 10% of particles are deposited and not transmitted, respectively."

**Specific comment 35:**

L343: What is the "original" transmission curve? Are there different versions of it?

Answer 35:

Thank you for your comment. We have removed "original" and now simply refer to "transmission curves."

Modification 35:

L343: "We analyzed the transmission curves for the ultrafine section of the particle size distribution for evaluating the losses of each impactor."

**Specific comment 36:**

L356: What is meant by "semi-industrial environment"? Is there heavy industry in the area? Do you expect strong UFP emissions nearby?

Answer 36:

We describe the sampling location as "semi-industrial" because the area has mixed use such as industrial activities, but no heavy industry, traffic, but no dense infrastructure. However, we understand that this term may lack clarity. Therefore, we have provided a more precise description of the sampling site in the manuscript to better convey the environmental context and potential sources of UFP.

Modification 36:

L 356: "…pumps were positioned indoors in the laboratories of the Bayreuth Center for Ecological and Environmental Research (BayCEER, 49.9305° N, 11.5881° E). The location is characterized by nearby manufacturing industries, road traffic, and proximity to a highway, but also adjacent to residential areas of the city of Bayreuth with around 75,000 inhabitants (Bayerisches Landesamt für Statistik, 2022)."

**Specific comment 37:**

L357: 30 L/min through 3/8" tubing should lead to turbulent air flow, I suspect. To avoid in-tube losses, laminar flow is typically preferred. Did the tubing add significant back-pressure to the setup? Did you check the sampling flow rate at the tubing inlet or at the impactor inlet (without the tubing)?

Answer 37:

The sampling flow rate was measured at the impactor inlet, and this procedure was consistently applied across all impactors. The line was kept as short as possible with 1.2m length. As such, we expect comparable effects across all devices. While we acknowledge that turbulent airflow may occur due to the tubing, we assume any resulting particle losses to have a more significant impact on larger particles, given their greater susceptibility to turbulence. In contrast, we expect ultrafine particles (UFPs) to be less affected by these conditions, and therefore the influence of turbulence on UFP measurements should be minimal in this setup.

**Specific comment 38:**

L362: Please report the O3 scrubbing efficiency.

Answer 38:

We have incorporated information on the ozone scrubbing capability into the text accordingly.

Modification 38:

L 361: "The ozone scrubbers were previously evaluated for their capacity to scrub ozone from the sampling air and for potential losses (≤ 6% for particles smaller than $d_m$=200 nm, ≤ 11% for particles larger than $d_m$=200 nm). Thereby, up to a concentration of 250 ppb ozone over 72 hours, ozone levels were reduced to below 5 ppb behind the ozone denuder (Eckenberger et al., in preparation)."

**Specific comment 39:**

L373: What temperature was used for filter baking?

Answer 39:

We have clarified that the filters were baked at 300°C for 24 hours prior to use.

Modification 39:

L374: "For collection of UFP, we inserted pre-baked quartz fiber filters (Whatman QM-H, 47mm or 37mm) in the after-filter holder of each impactor, pre-baked at 300°C for 24h. These were stored at -20°C immediately after collection."

**Specific comment 40:**

L391: Custom-made glass frits or commercial ones? Why not use conventional syringe filters instead?

Answer 40:

We considered using the samples for potential future studies involving microplastics. Therefore, filtration using commercially available syringe filters, which are made of plastic, was excluded to avoid contamination and ensure compatibility with microplastic analysis. Furthermore, we found the glass frits to be a sustainable solution avoiding one-way syringe filters. Their cleanliness was tested repeatedly with running blanks along the entire extraction procedure.

**Specific comment 41+42:**

L397: Can you estimate the volume of the residual droplet?

L398 and elsewhere: "Millipore water" does not really fit, as your system was not a Millipore system.

Answer 41+42:

We have added the estimated volume of the residual droplet to the text and have removed "Millipore" from the text as suggested.

Modification 41+42:

L396-398: "A droplet was kept as residue, approximately 0.5 µL in volume, which was dissolved in 1 mL of a 60:40 solution of Acetonitrile (ACN, Carl Roth, 99.95%) and water ($H_2O$, obtained from Seralpur PRO 90 CN system with Supor DCF filter, Electronics Grade, 0.2 µm)."

L385: "For the mobile phase, HPLC-grade acetonitrile, water, and formic acid as buffer (HCOOH, Carl Roth, p.a. ≥ 98%) was used."

**Specific comment 43+44:**

L410: Which reference material?

L428: Which NIST standard? Where and how was this used, it has not been mentioned before.

Answer 43+44:

We have specified the reference material as "NIST SRM 2786 with PM < 4 µm" in the text. We have included a reference to the NIST standard and a description of procedure in the text.

Modification 43+44:

L428: "The recovery-corrected results were then validated against the NIST standard (SRM 2786, <4 µm) which was applied to a filter, extracted and analysed according to the methods described herein, for validation and showed agreement within the uncertainty of the measurement, even in the presence of a particulate matrix."

**Specific comment 45:**

L431: Please give the IC LODs in the Supplement Table as well, not only the air-equivalent ones.

Answer 45:

We have added the IC LODs to the Supplement Table, alongside the air-equivalent LODs, as requested. Please, find it in the SI accordingly.

**Specific comment 46:**

L436ff: There might be an error here. The given sampling volumes refer to the entire filter, while only half of the filter was actually extracted (L415). This needs to be included in the air-equivalent LOD calculation and the given LODs might be different by a factor of 2. Please check.

Answer 46:

We have taken the entire filter into account during the air-equivalent LOD calculation by multiplying the values by a factor of 2, ensuring that the calculations reflect the total sampling volume. This adjustment aligns the LOD with the overall measurement volume, so there is no discrepancy.

**Specific comment 47:**

L441: Better explain the different considered uncertainties here (briefly) and give a Table of their values in the Supplement (rather than text).

Answer 47:

We have revised the text to provide a clearer and more concise explanation of the different uncertainties considered in our analysis. Additionally, as requested, we have added a table summarizing these uncertainty values in the Supplement to improve clarity and accessibility for the reader (Table S2).

Modification 47:

L441: "..with uncertainties ranging from 13.8% to 17.8% (SI 1.1). This estimate for the uncertainty includes the instrumental variability for repeated analysis (2%), reference material uncertainty (0.5%), preparations of stock solutions (5%), dilution errors (5%), inaccuracies in volume determination (1%). Additionally, it accounts for impactor-specific factors, including flow adjustment errors (3–10%) and handling errors (10–15%)."

**Specific comment 48:**

L456 and Table 1: Aerodynamic cut-off diameters should be calculated not only for NaCl, but also for SimSOA and soot, using appropriate values for density and shape factor. Only these can be compared to the nominal impactor cut-points, which are defined in aerodynamic diameter as well.

Answer 48:

Thank you for the suggestion. We have calculated the aerodynamic cut-off diameter for SimSOA and included it in Table 1. However, for soot, we chose not to calculate the aerodynamic cut-off diameter due to significant uncertainty regarding both the shape factor and density of soot particles. Soot's irregular morphology and agglomerated structure result in a wide variation in these parameters, making it difficult to reliably estimate an aerodynamic diameter for comparison with nominal impactor cut-points. Therefore, we focused on NaCl and SimSOA, where the shape factor and density could be reasonably approximated.

Modifications 48:

L457: "For NaCl and SimSOA particles, we compared the aerodynamic cut-off diameters, which were 86±9 and 84±8 (ultraMOUDI), 89±9 and 85±9 (120R MOUDI), 99±10 and 89±9 (PENS), and 100±10 and 85±9 (ELPI) nm for NaCl and SimSOA, respectively. For the calculation, a shape factor of 1.0 and a density of 2.165 g/cm³ were used for NaCl, while a shape factor of 1.2 and a density of 1.21 g/cm³ were applied for SimSOA."

**Specific comment 49:**

L466: See comment above on dp10 and dp90.

Answer 49:

Thank you, we have reviewed it again and clarified the description in the text accordingly.

**Specific comment 50:**

L469: Unclear and possibly misleading statement, please check. What do you mean by "lower efficiency for capturing small particles"? Did you consider the differences between aerodynamic and electromobility diameter here?

Answer 50:

Thank you for this comment. As we now included the aerodynamic diameter calculation for SimSOA, we discussed the cut-off diameter later in the manuscript. At this point, we have revised the text to clarify the description of dp10 values without implying differences in collection efficiency.

Modification 50:

L456: "For all impactors, the lowest dp10 values were observed with NaCl particles ($dp10_{ave}$=34 nm, average across all impactors)."

**Specific comment 51:**

L472: The stickiness of NaCl depends on RH and their water shell. Not sure they can generally be considered sticker than SOA, which often is semi-liquid and thus rather sticky.

Answer 51:

We have adjusted the wording to reflect that NaCl and SimSOA have similar stickiness, rather than implying that NaCl is generally stickier.

Modification 51:

L472-475: "As this general trend might be driven by the relatively sticky nature of NaCl particles and SimSOA leading to a relatively smeared cut-off, it is interesting to observe the sharpest cut-off for the soot particles, which exhibit the most complex shape of the three tested particle types."

**Specific comment 52:**

L481ff: Unclear paragraph. What is a "suitable" cut-off? Are slight deviations from 100 nm really relevant, especially when given in a different diameter (electromobility) and for a mass-based analysis? Unclear, how MOUDIs would "shift" their cut-off diameter in "more realistic air mixtures", as the cut-point is actually fixed by design. Please, rephrase entire paragraph.

Answer 52:

We have revised this section to reduce subjective language and clarify the statements regarding cut-off behavior.

Modification 52:

L481-486: "The estimated aerodynamic cut-off diameters for SimSOA and NaCl overlap for all the impactors within their uncertainties in a range of 84-100 nm. Typically, the UFP range includes the nucleation mode and marks the shoulder of the accumulation mode. With the perspective of a mass based analysis, particle composition of these two modes would be reflected in the results. Yet, as the mass of particles around 100 nm is small, the observed variations in cut-off diameter likely do not diminish their comparability."

**Specific comment 53:**

Table 1: The figures of the impactors are misleading, as the Table characterizes their last stage only (or close-to-last, if modified). They could be removed to make the Table more compact.

Answer 53:

We have removed the figures from Table 1 to make it more compact, as suggested.

**Specific comment 54:**

Figure 3, caption: What does "original impactors" mean? Were these experiments performed in the original configuration of MOUDI and ELPI, i.e. without removing last stages?

Answer 54:

We have updated the caption to simply state that all tests were performed without coating.

Modification 54:

"now Figure 4: Transmission curves for the four tested impactors collecting SimSOA generated within the atmospheric simulation chamber. Note that all tests were performed without applying coating. The four impactors are marked as 120R MOUDI (orange, circle), PENS (purple, triangle), ELPI (green, square), and ultraMOUDI (blue, diamond markers)."

**Specific comment 55:**

L520: Transmission of the ELPI is quite visibly increasing at largest diameters. I doubt, the statement is justified. It is also unclear, which "uncertainty" the authors refer to here. A transmission of 6% is likely significantly larger than 0, depending on the total transmission concentration.

Answer 55:

We acknowledge the slight increase in transmission for the largest particle diameters observed in the ELPI. However, this is primarily due to the fact that very few particles were present at

these diameters. With such low particle counts, even a single particle can have a significant impact on the calculated transmission ratio. This means that the transmission values at these larger diameters should be interpreted with caution. Despite this, the observed values still lie within the overall uncertainty of the measurement, given the small number of particles detected at these sizes. We have adjusted the text accordingly.

Modification 55:

L519: "The ELPI detected very few particles at the larger diameters, with minimal transmission observed for these particle sizes."

**Specific comment 56:**

L521: Similarly, the MOUDI line in Figure 3 does not seem to exactly reach 0, i.e. "point of no transmission".

Answer 56:

We have revised the text to clarify that very few particles were detected at larger diameters, and the transmission values reflect this, similar to the adjustments made for the ELPI.

Modification 56:

L520-522: "A similar observation can be made for the 120R MOUDI, where very few particles were detected at larger diameters, approaching minimal transmission at an electromobility diameter of 562 nm."

**Specific comment 57:**

L534: Losses for the MOUDI were basically the same as for the ultraMOUDI. For ELPI, they were just slightly higher (L505). The statement of "increased losses" is not exactly correct. Also, slightly higher losses likely have much less impact on final mass-based concentrations as bounce-off of large particles.

Answer 57:

We have revised the text to correct the statement regarding losses and its impact

Modification 57:

L539-557: :" Here, the losses in the UFP range were similar for ELPI, 120R MOUDI and ultraMOUDI. This contrasts with the PENS, which had the smallest loss amongst all tested models with about 6%, likely due to its design with the cyclone pre-separator and only one nozzle plate. We can compare our results to a few other studies, that experimentally determined the loss rate of the MOUDI. Liu et al. (2013) showed a total loss of a MOUDI (Model 110) in the range of 2.9-26.1% increasing with decreasing dp50, which was attributed mostly to convective-diffusion. Similarly, Durand et al., (2014) observed losses by convection-diffusion

in cascade impactors with stages designed for ultrafine particles below 100 nm. Ungeheuer et al. (2022) measured losses of 28% and 40% in the Nano-MOUDI (110) for particles with aerodynamic diameter of 32-56 nm and 18-32 nm, respectively. It is thought that diffusion deposition becomes increasingly significant for smaller particles, which can lead to substantial particle losses. As high uncertainties are associated with both MPSS and DMS, when measuring particles with diameters below 20 nm, we could not test this behaviour within out setup. However, a mass based analysis of UFP might be less affected by such losses than the measurement of the number concentration.

It has been reported, that particle bounce becomes particularly significant for lower cut-off stages because of the gradual reduction in pressure at each stage, which subsequently leads to a decrease in relative humidity (RH). The reduction in RH in turn can intensify the particle bounce effect (Chen et al., 2011). Pressure dropped most drastically throughout the ELPI, which could be problematic for collecting semi-volatile organic marker compounds (Yao et al., 2022). Knowing that the diameters of cut-off and the effective sharpness of the separation between fine and ultrafine SimSOA particles were comparable for all tested impactors, the impact of losses and particle bounce on a mass based chemical analysis might be significant, which therefore is further investigated in the following."

**Specific comment 58:**

L592ff and Fig. 5: These data should be used to estimate the potential mass bias from bounce-off. Stating that 98.5% of particle mass was captured might be misleading, if the actual UFP mass was much smaller than 1.5%. From Fig. 3, it seems the UFP mass was ~0.7 µg/m3. If transmission to the UFP stage was indeed 3.3 µg/m3, this would mean that only ~20% of UFP mass concentration is actually from the ultrafine particle size range, while 80% of mass (and chemical composition) are bias from the bounce-off sampling artefact. Marker concentrations from such biased sampling would be difficult to trust. If such data is available for the other impactors as well, similar estimates should be done and reported. Bounce-off is a well-known and critical artefact of impactor sampling and should be characterized as quantitatively as possible from these lab experiments. It might also be worth mentioning that SOA is likely not a worst-case scenario for bounce-off, given the comparably sticky nature of SOA particles.

Answer 58:

This is a valuable comment. Firstly, we would like to clarify that the percentages refer to the total particle mass across the 10–800 nm range. From this perspective, we believe the statement is accurate that 98.5% of the particle mass was captured by the impactor when coating was applied compared to only 90.3% without the coating. The figure and discussion on the bounce effect further illustrate one of the primary challenges in the separation and collection of UFP for intentional mass-based chemical characterization, specifically how this mass can be significantly influenced by various effects.

Secondly, we agree with the reviewer that it would be very interesting to make a similar statement for the other impactors and include them in the comparison. However, these

measurements were unfortunately limited. In order to ensure a relatively fair measurement in the context of environmental measurements, we aimed to keep the particle number concentration as low as possible, since, as discussed in the paper, the extent of the bounce effect is dependent on particle number concentration. Due to this lower number concentration, the DMS 500 was no longer a suitable instrument, as it has a relatively high detection limit. Therefore, we switched to a more sensitive MPSS. However, since the MPSS is highly sensitive to pressure drops before its inlet, measurements could only be conducted with the impactor that had the lowest pressure drop, the PENS.

Thirdly, we see that the wording of the paragraph is misleading and not very clear. We revised the text in order to improve and sharpen its message. Furthermore, the statement regarding the worst-case scenario has been incorporated accordingly.

Modification 58:

L593-L597: "The set of measurements includes particle mass as measured directly from the chamber, from the empty PENS, the uncoated PENS, and the coated PENS (Fig. 5). The total mass of all particles in the chamber ranging from 10 to 800 nm, calculated directly from the measured particle number size distribution, was 220.2 µg/m³. Due to losses in the lines and the empty PENS, the mass was reduced to 205.87 µg/m³. Upon separation, 21.4 µg/m³ were transmitted through the PENS when no coating was applied. Through the coated PENS a particle mass of 3.3 µg/m³ was transmitted. This highlights the coated impactor's capability to retain over 98.5% of the test particle mass, whereas the uncoated impactor captured only approximately 90.3%. The difference are artefacts due to bounce and fragmentation of larger particles sampled as UFP. Figure 5 shows also, that the transmitted ultrafine fraction as detected with MPSS remained comparable between coated and uncoated tests. For this test, we chose a reduced particle number concentration, compared to the above cut-off characterizing experiments, and SimSOA particles in the context of environmental measurements. Due to the nature of SimSOA they are likely not representative of a worst-case scenario for particle bounce."

**Specific comment 59:**

Section 3.4: Page 25 largely repeats what has been given in the experimental section already. Please make sure to remove redundancies throughout the entire manuscript.

Answer 59:

Thank you for the feedback. We will review the manuscript carefully to identify and remove redundancies, particularly on Page 25 and throughout the experimental section, to improve clarity and conciseness.

**Specific comment 60:**

L641: Better "deviation" instead of "over- or underestimation" as there is no ground truth available.

Answer 60:

We have replaced "over- or underestimation" with "deviation," as suggested.

Modification 60:

L640-L643: "We analyzed UFP sampled in an urban, semi-industrial environment. The observed average mass concentrations are presented in Figure 7A. Furthermore, we evaluated the relative deviation of each impactor compared to the average of all impactor results (now Figure 7B)."

**Specific comment 61:**

L648: 37% maximum difference seems low (from Fig. 6), please double-check.

Answer 61:

Thank you for pointing this out. We have rechecked the calculations and corrected the value accordingly.

Modification 61:

L655: "We can highlight two findings here for the analysis of the two PAHs: Firstly, the maximum disagreement can be as large as 59% between PENS and 120R MOUDI."

**Specific comment 62:**

L648: How do you know about >90% particle phase fraction of PAHs?

Answer 62:

Benzo[a]pyrene (BaP) and Benzo[b]fluoranthene (BbF) are high molecular weight polycyclic aromatic hydrocarbons (PAHs) with low volatility, leading them to predominantly exist in the particle phase under ambient conditions. This behavior has been documented in various environmental studies. For instance, (Maharaj Kumari and Lakhani, 2018; Manoli et al., 2016). These references have been included now.

**Specific comment 63:**

L689-690: Given that with the impactors applied, the potential of evaporative losses is inversely related to the potential of bounce-off artifacts, I am not convinced such a statement can be made. From Fig. 7A it can be seen that between MOUDI and ELPI, marker concentrations often increase, even though the ELPI has the largest pressure drop. This could point to bounce-off having a stronger impact on marker concentrations than evaporation. Potential evaporative losses should better be discussed based on the volatilities of the marker compounds, possibly in combination with the pressure drops experienced.

Answer 63:

We agree that the trends observed in the data, particularly for BaP, BbF, and 6PPD, suggest that factors beyond pressure drop alone may influence marker concentrations. This complexity is indeed an important consideration when interpreting the results. To address this, we made modifications in our manuscript (L736-744) to provide a more nuanced analysis. Specifically, we categorized the compounds into two groups based on their expected volatility: semi-volatile organic acids (Pinic Acid and Terpenylic Acid) and less volatile PAHs (BaP and BbF). We quantified the impact of pressure drop on the absolute concentration. Additionally, we reinforced our statement regarding the potential influence of other effects that may be more significant than the pressure drop, acknowledging that these factors could have a pronounced impact on marker concentrations.

Modification 63:

L680-L691: "We expected the pressure drop to primarily affect the semi-volatile markers, PA, TA, and Levo, due to evaporation losses during sampling. Figure 7a illustrates the dependency of the absolute deviation of the average mass concentration on the pressure drop for all samples and impactors. The absolute deviation of the average mass concentrations of PA, TA, and Levo decreased with increasing pressure drop, stronger than the less volatile markers, BaP and BbF. We lumped these two groups of markers, semi-volatile and mostly particle-bound, and compared the change in concentration for the two extremes, the PENS with the smallest pressure drop (260±1 hPa) and the ELPI with the largest pressure drop (690±3 hPa). The lower-volatility markers (BaP,Bbf) decreased in average concentration about 15%. In contrast, the average mass concentration of the higher-volatility markers (PA, TA) decreased about 52%. These findings seem to confirm that the larger pressure drop in the impactors leads to evaporation and thus a mass loss of the semi-volatile substances. Likely, the evaporative loss depends also on the collection substrate. When sampled in QFF, for example, UFP are exposed individually and ventilated efficiently. However, here QFF was used in all impactors for UFP sampling. Furthermore, we firstly noted differences between 120R MOUDI and ELPI, despite of a comparable pressure drop. Secondly, mass concentrations of 6PPD were comparable for all impactors except for the PENS. These observations indicate additional influencing factors beyond the pressure drop alone, which could potentially affect the analysis more significantly."

**Specific comment 64:**

L698: Not sure if I follow here. How would the PENS attribute twice as much mass to UFP as there is total ambient mass available? And for the MOUDI it would be 18% of the total ambient PM mass? There is quite large scatter in Fig. 7B, and on logarithmic axes. Please double-check if these statements are regarded robust.

Answer 64:

Thank you for pointing this out. We revisited our analysis and carefully reviewed the data in now Fig. 8B, especially considering the scatter and the use of logarithmic axes. We acknowledge that the observed scatter complicates the interpretation and introduces

uncertainty. However, we would like to focus our analysis on identifying potential patterns related to particle bounce. Thus, we modified Figure 7B ( now 8B) and rewrote the respective paragraph accordingly.

Modification 64:

L693-700: "Despite the application of a coating, we suspected particle bounce and thus tested whether the absolute deviation from the impactor average mass concentration was dependent on the ambient marker mass concentration (Fig. 7b). In case of particle bounce, we expected an increasing effect with increasing mass concentration. Yet, any potential dependency could only be seen in our data for the highest daily average concentrations, particularly above 1 ng/m³. Below this level, the data do not suggest any systematic influence of particle bounce."

**Specific comment 65:**

L740: transmission of large particles is not entirely eliminated.

Answer 65:

We have revised the sentence to clarify that the transmission of large particles is largely reduced.

Modification 65:

L757: "We showed that the application of a coating largely reduced the transmission of larger particles in the laboratory tests."

**Specific comment 66:**

L750: The loss is not 69%.

Answer 66:

Thank you for pointing that out. We agree, and have corrected the value accordingly in the text.

Modification 66:

L749-750: "However, this comes with an overall loss in the ultrafine fraction (31±7%) and a pressure drop of 690 hPa. The reduced pressure on the filter substrate can lead to the evaporation of semi-volatile substances during sampling."

**Specific comment 67:**

L752: Considering the confounding with potential bounce-off artefacts, the concentration differences between ELPI and ultraMOUDI are not high enough to dismiss its suitability for semi-volatile UFP components, in my opinion. The same applies to the MOUDI. Not considered at all in this study are possible differences of evaporative losses from quartz fibre filters

(particles deposited individually on fibres) as compared to the more typical flat substrates (particles deposited in dense spots or layers).

Answer 67:

We have revised the concluding text parts for each impactor to present the information in a more neutral and non-evaluative manner. The discussion of evaporative losses from QFF has been included earlier in the text. Please refer to Modification 63 as well.

Modification 67 (accordingly for the other impactors):

L749-754: "However, this comes with an overall loss in the ultrafine fraction (31±7%) and a pressure drop of 690 hPa. The reduced pressure on the filter substrate can lead evaporative loss of semi-volatile substances during sampling, while non-volatile, particle-bound compounds in UFP (e.g. metals, plastics) can be unaffected by this effect."

**Specific comment 68:**

L752: What does "stable" mean in this context?

Answer 68:

We have clarified that "stable" in this context refers to non-volatile compounds.

Modifcation 68:

L752: "The reduced pressure on the filter substrate can lead evaporative loss of semi-volatile substances during sampling, while non-volatile, particle-bound compounds in UFP (e.g. metals, plastics) can be unaffected by this effect."

**Specific comment 69:**

L760: If 690 hPa (ELPI) is "high" and 420 hPa (ultraMOUDI) is "moderate", I would consider 600 hPa as rather high as well.

Answer 69:

Thank you for this comment. We have adjusted the relevant sections in the text accordingly.

Modification 69:

L760: ". The 120R MOUDI has a pressure drop of 600 hPa. Thus, for semi-volatile substances evaporative losses are likely. "

References

Bayerisches Landesamt für Statistik: Statistik kommunal - Kreisfreie Stadt Bayreuth - 09 462, 1–30 pp., 2022.

Bein, K. J. and Wexler, A. S.: A high-efficiency, low-bias method for extracting particulate matter from filter and impactor substrates, Atmos Environ, 90, 87–95, 2014.

Canepari, S., Astolfi, M. L., Moretti, S., and Curini, R.: Comparison of extracting solutions for elemental fractionation in airborne particulate matter, Talanta, 82, 834–844, https://doi.org/10.1016/j.talanta.2010.05.068, 2010.

Chien, C. L., Tien, C. Y., Liu, C. N., Ye, H., Huang, W., and Tsai, C. J.: Design and Testing of the NCTU Micro-Orifice Cascade Impactor (NMCI) for the Measurement of Nanoparticle Size Distributions, Aerosol Science and Technology, 49, 1009–1018, https://doi.org/10.1080/02786826.2015.1089976, 2015.

Crazzolara, C. and Held, A.: Development of a cascade impactor optimized for size-fractionated analysis of aerosol metal content by total reflection X-ray fluorescence spectroscopy (TXRF), Atmospheric Measurement Techniques, 17, 2183–2194, 2024.

Daher, N., Ning, Z., Cho, A. K., Shafer, M., Schauer, J. J., and Sioutas, C.: Comparison of the chemical and oxidative characteristics of particulate matter (PM) collected by different methods: Filters, impactors, and BioSamplers, Aerosol Science and Technology, 45, 1294–1304, https://doi.org/10.1080/02786826.2011.590554, 2011.

Fujitani, Y., Hasegawa, S., Fushimi, A., and Kondo, Y.: Collection characteristics of low-pressure impactors with various impaction substrate materials, Atmos Environ, 40, 3221–3229, 2006a.

Fujitani, Y., Hasegawa, S., Fushimi, A., Kondo, Y., Tanabe, K., Kobayashi, S., and Kobayashi, T.: Collection characteristics of low-pressure impactors with various impaction substrate materials, Atmos Environ, 40, 3221–3229, https://doi.org/10.1016/j.atmosenv.2006.02.001, 2006b.

Järvinen, A., Aitomaa, M., Rostedt, A., Keskinen, J., and Yli-Ojanperä, J.: Calibration of the new electrical low pressure impactor (ELPI+), J Aerosol Sci, 69, 150–159, https://doi.org/10.1016/j.jaerosci.2013.12.006, 2014.

Kim, S., Shen, S., Sioutas, C., Zhu, Y., and Hinds, W. C.: Size distribution and diurnal and seasonal trends of ultrafine particles in source and receptor sites of the Los Angeles Basin, J Air Waste Manage Assoc, 52, 297–307, https://doi.org/10.1080/10473289.2002.10470781, 2002.

Kittelson, D., Khalek, I., McDonald, J., Stevens, J., and Giannelli, R.: Particle emissions from mobile sources: Discussion of ultrafine particle emissions and definition, https://doi.org/10.1016/j.jaerosci.2021.105881, 1 January 2022.

Löber, M., Bondorf, L., Grein, T., Reiland, S., Wieser, S., Epple, F., Philipps, F., and Schripp, T.: Investigations of airborne tire and brake wear particles using a novel vehicle design, Environmental Science and Pollution Research, 31, 53521–53531, https://doi.org/10.1007/s11356-024-34543-9, 2024.

Maharaj Kumari, K. and Lakhani, A.: PAHs in Gas and Particulate Phases: Measurement and Control, in: Environmental Contaminants: Measurement, Modelling and Control, edited by: Gupta, T., Agarwal, A.

K., Agarwal, R. A., and Labhsetwar, N. K., Springer Singapore, Singapore, 43–75, https://doi.org/10.1007/978-981-10-7332-8_3, 2018.

Manoli, E., Kouras, A., Karagkiozidou, O., Argyropoulos, G., Voutsa, D., and Samara, C.: Polycyclic aromatic hydrocarbons (PAHs) at traffic and urban background sites of northern Greece: source apportionment of ambient PAH levels and PAH-induced lung cancer risk, Environmental Science and Pollution Research, 23, 3556–3568, https://doi.org/10.1007/s11356-015-5573-5, 2016.

Marple, V., Olson, B., Romay, F., Hudak, G., Geerts, S. M., and Lundgren, D.: Second generation micro-orifice uniform deposit impactor, 120 MOUDI-II: Design, Evaluation, and application to long-term ambient sampling, Aerosol Science and Technology, 48, 427–433, https://doi.org/10.1080/02786826.2014.884274, 2014.

Mazzi, G., Feltracco, M., Barbaro, E., Alterio, A., Favaro, E., Azri, C., and Gambaro, A.: Glyphosate and other plant protection products in size-segregated urban aerosol: Occurrence and dimensional trend, Environmental Pollution, 359, https://doi.org/10.1016/j.envpol.2024.124596, 2024.

Ngagine, S. H., Deboudt, K., Flament, P., Choël, M., Kulinski, P., and Marteel, F.: Development and Characterization of a Time-Sequenced Cascade Impactor: Application to Transient PM2.5 Pollution Events in Urbanized and Industrialized Environments, Atmosphere (Basel), 13, https://doi.org/10.3390/atmos13020244, 2022.

Romay, F. J. and García-Ruiz, E.: Design of Round-Nozzle Inertial Impactors Review with Updated Design Parameters, Aerosol Air Qual Res, 23, https://doi.org/10.4209/aaqr.220436, 2023.

Tsai, C. J., Liu, C. N., Hung, S. M., Chen, S. C., Uang, S. N., Cheng, Y. S., and Zhou, Y.: Novel active personal nanoparticle sampler for the exposure assessment of nanoparticles in workplaces, Environ Sci Technol, 46, 4546–4552, https://doi.org/10.1021/es204580f, 2012.

Wang, D., Jia, S., Liu, L., and Zhang: Pollution characteristics, source apportionment and absorption spectra of size-resolved PAHs in atmospheric particles in a cold megacity of China, J Hazard Mater, 43, 2024.

---

## Author Comment (AC2)

We appreciate the reviewer's careful evaluation of our manuscript, "Performance evaluation of four cascade impactors for airborne UFP collection: Influence of particle type, concentration, mass, and chemical nature." The feedback has been very helpful in identifying areas where the manuscript could be clarified and refined. We have addressed each of the reviewer's comments in detail and have made corresponding revisions to improve the overall quality and focus of the paper. Our responses, along with specific modifications made to the text, are provided below.

This study inter-compares different cascade impactors able to sample UFPs. As the earlier devices based on particle collection by inertial impaction did not have a good size-resolution below 0.1 um, different modifications/extensions of the original impactors were performed. These modified versions implying different underlying principles and design brought a variety of advantages and disadvantages, strengths and weaknesses. Therefore, it is an original idea and a useful task to compare them. It is one of the strengths of this manuscript that the selected devices were compared both under laboratory and field conditions. It is also remarkable that the authors used different test aerosol systems with different properties allowing for the study of evaporation and bouncing, two confounding factors of precise aerosol size distribution measurement. Another strength of the work is the exigent planning, completion and analysis of the related experiments, as well as a precise description of their work and presentation of the results both in the manuscript and in the supplementary material.

**Comment 1:**

> In the light of the use of modified impactors instead of the original ones an imminent question is the relevance of current results regarding the original impactors. The authors should include a section or paragraph commenting on this important issue.

Answer 1:

Thank you for bringing up this important point regarding the relevance of our results to the original impactors. We agree that discussing the implications of using modified impactors is crucial. In response, we have included sections in the manuscript where we comment on the potential impacts of these modifications. We have also clarified these points more thoroughly in the abstract, methods, and throughout the manuscript to ensure that the relevance of our findings to the original impactors is clearly communicated.

Modifications 1:

L148-151: "Today, several cascade impactors exist, which are either commercially available or newly developed (Crazzolara and Held, 2024; Ngagine et al., 2022; Järvinen et al., 2014; Romay and García-Ruiz, 2023; Marple et al., 2014a; Tsai et al., 2012b). For our comparison, we selected four commercially available models for sampling atmospheric UFP that cover different designs, flow rates, and stage numbers. Additionally, we wanted to sample all ≤100 nm particles on one substrate without further separation. Moreover, we envisioned the use of an automated filter changer in future applications which would be possible with all selected models. Some of these selected impactors required minor adjustments to make them suitable for achieving the final cut-off diameter at 100 nm. Apart from these adjustments, which are outlined in the following, the cascade impactors were operated as described by the manufacturers"

L165: "In this study, we modified the 120R MOUDI by removing the 0.56 nm stage (including the nozzle and impaction plates) located below the 100 nm cut-off diameter stage. This modification allowed us to collect all particles ≤100 nm in the original after-filter holder mounted at the impactor outlet."

L193-197: "For this study, we extracted the cascade impactor component from the ELPI and considered it as a standalone impactor without the charger and electrometer. For the collection of UFP, we removed the stages with cut-off diameters of 0.03 and 0.06 μm to achieve a final cut-off size of 0.09 μm at stage 3. To maintain the flow characteristics, secure the impaction plates in the built-in tensioner, and ensure appropriate spacing between the nozzle and collection plates, placeholders were inserted instead. On upper stages, aluminium foil filters (25mm, Dekati) were installed. For collection of UFP, a 37 mm QFF was installed in the after-filter holder provided by the manufacturer."

**Comment 2:**

In addition, the authors did not evaluate losses for particles with diameter below 30 nm. They mentioned that „Due to the relatively larger uncertainties in the reference instruments for very small diameters, i.e. $d_m$ <20 nm, we decided to evaluate the particle number concentration at 30 nm for determining the losses in the ultrafine fraction". As deposition by diffusion increases steeply below this size, it would be important to reflect on this issue, at least by expert judgement or/and using data from the open literature.

Answer 2:

We sincerely thank the reviewer for highlighting this important point. To address this concern and provide greater clarity, we have revised the text accordingly. Specifically, we have emphasized the rationale for starting our analysis at 30 nm, given the high uncertainties associated with measuring very small particle diameters and the significant impact of diffusion deposition below this threshold. Additionally, we have incorporated relevant literature on particle losses in cascade impactors, as suggested, to support our discussion. We have also clarified how the focus on mass-based chemical analysis in our study mitigates the relative influence of such diffusion losses.

Modification 2:

L539-557: "Here, the losses in the UFP range were similar for ELPI, 120R MOUDI and ultraMOUDI. This contrasts with the PENS, which had the smallest loss amongst all tested models with about 6%, likely due to its design with the cyclone pre-separator and only one nozzle plate. We can compare our results to a few other studies, that experimentally determined the loss rate of the MOUDI. Liu et al. (2013) showed a total loss of a MOUDI (Model 110) in the range of 2.9-26.1% increasing with decreasing $dp50$, which was attributed mostly to convective-diffusion. Similarly, Durand et al., (2014) observed losses by convection-diffusion in cascade impactors with stages designed for ultrafine particles below 100 nm. Ungeheuer et al. (2022) measured losses of 28% and 40% in the Nano-MOUDI (110) for particles with aerodynamic diameter of 32-56 nm and 18-32 nm, respectively. It is thought that diffusion deposition becomes increasingly significant for smaller particles, which can lead to substantial particle losses. As high uncertainties are associated with both MPSS and DMS, when measuring particles with diameters below 20 nm, we could not test this behaviour within out setup. However, a mass based analysis of UFP might be less affected by suchh losses than the measurement of the number concentration.

It has been reported, that particle bounce becomes particularly significant for lower cut-off stages because of the gradual reduction in pressure at each stage, which subsequently leads to a decrease in relative humidity (RH). The reduction in RH in turn can intensify the particle bounce effect (Chen et al., 2011). Pressure dropped most drastically throughout the ELPI, which could be problematic for collecting semi-volatile organic marker compounds (Yao et al., 2022). Knowing that the diameters of cut-off and the effective sharpness of the separation between fine and ultrafine SimSOA particles were comparable for all tested impactors, the impact of losses and particle bounce on a mass based chemical analysis might be significant, which therefore is further investigated in the following."

Comment 3:

Finally, the manuscript is quite long. If the authors can find a way to compact it without loss of pertinent information, it would be nice. For instance, the introduction could be shortened, but I would let the authors to decide on what to shorten.

Answer 3:

Thank you for your comment regarding the length of the manuscript. In response, we have made adjustments to reduce its overall length. Specifically, we have shortened the introduction and streamlined the description of the measurement methods. Please find these amendments throughout the revised manuscript.